# Signatory: differentiable computations of the signature and logsignature transforms, on both CPU and GPU

**Patrick Kidger**     **Terry Lyons**
Mathematical Institute, University of Oxford
The Alan Turing Institute, British Library
`{kidger, tlyons}@maths.ox.ac.uk`

## ABSTRACT

Signatory is a library for calculating and performing functionality related to the signature and logsignature transforms. The focus is on machine learning, and as such includes features such as CPU parallelism, GPU support, and backpropagation. To our knowledge it is the first GPU-capable library for these operations. Signatory implements new features not available in previous libraries, such as efficient precomputation strategies. Furthermore, several novel algorithmic improvements are introduced, producing substantial real-world speedups even on the CPU without parallelism. The library operates as a Python wrapper around C++, and is compatible with the PyTorch ecosystem. It may be installed directly via `pip`. Source code, documentation, examples, benchmarks and tests may be found at `https://github.com/patrick-kidger/signatory`. The license is Apache-2.0.

## 1 INTRODUCTION

The *signature transform*, sometimes referred to as the *path signature* or simply *signature*, is a central object in rough path theory (Lyons, 1998; 2014). It is a transformation on differentiable paths[1], and may be thought of as loosely analogous to the Fourier transform. However whilst the Fourier transform extracts information about frequency, treats each channel separately, and is linear, the signature transform exacts information about order and area, explicitly considers combinations of channels, and is in a precise sense 'universally nonlinear' (Bonnier et al., 2019, Proposition A.6).

The *logsignature transform* (Liao et al., 2019) is a related transform, that we will also consider. In both cases, by treating sequences of data as continuous paths, then the (log)signature transform may be applied for use in problems with sequential structure, such as time series. Indeed there is a significant body of work using the (log)signature transform in machine learning, with examples ranging from handwriting identification to sepsis prediction, see for example Morrill et al. (2019); Fermanian (2019); Király & Oberhauser (2019); Toth & Oberhauser (2020); Morrill et al. (2020b).

Earlier work often used the signature and logsignature transforms as a feature transformation. See Levin et al. (2013); Chevyrev & Kormilitzin (2016); Yang et al. (2016a;b); Kormilitzin et al. (2016); Li et al. (2017); Perez Arribas et al. (2018) for a range of examples. In this context, when training a model on top, it is sufficent to simply preprocess the entire dataset with the signature or logsignature transform, and then save the result.

However, recent work has focused on embedding the signature and logsignature transforms within neural networks. Recent work includes Bonnier et al. (2019); Liao et al. (2019); Moor et al. (2020); Morrill et al. (2020a); Kidger et al. (2020) among others. In this context, the signature and logsignature transforms are evaluated many times throughout a training procedure, and as such efficient and differentiable implementations are crucial. Previous libraries (Lyons, 2017; Reizenstein & Graham, 2018) have been CPU-only and single-threaded, and quickly become the major source of slowdown when training and evaluating these networks.

---

[1]And may be extended to paths of bounded variation, or merely finite $p$-variation (Lyons et al., 2004).

## 1.1 CONTRIBUTIONS

We introduce Signatory, a CPU- and GPU-capable library for calculating and performing functionality related to the signature and logsignature transforms. To our knowledge it is the first GPU-capable library for these operations. The focus is on machine learning applications.

Signatory is significantly faster than previous libraries (whether run on the CPU or the GPU), due to a combination of parallelism and novel algorithmic improvements. In particular the latter includes both uniform and asymptotic rate improvements over previous algorithms. Additionally, Signatory provides functionality not available in previous libraries, such as precomputation strategies for efficient querying of the (log)signature transform over arbitrary overlapping intervals.

The library integrates with the open source PyTorch ecosystem and runs on Linux or Windows. Documentation, examples, benchmarks and tests form a part of the project.

Much of the code is written in C++ primitives and the CPU implementation utilises OpenMP. The backward operations are handwritten for both speed and memory efficiency, and do not rely on the autodifferentiation provided by PyTorch.

The source code is located at `https://github.com/patrick-kidger/signatory`, documentation and examples are available at `https://signatory.readthedocs.io`, and the project may be installed directly via `pip`.

This paper is not a guide to using Signatory—for that we refer to the documentation. This is meant as a technical exposition of its innovations.

## 1.2 APPLICATIONS

Signatory has already seen a rapid uptake amongst the signature community. Recent work using Signatory include Morrill et al. (2020b); Perez Arribas et al. (2020) who involve signatures in neural differential equations, or Moor et al. (2020); Min & Ichiba (2020) who study deep signature models (Bonnier et al., 2019). Meanwhile Ni et al. (2020) apply Signatory to hybridise signatures with GANs, and Morrill et al. (2020a) create a generalised framework for the "signature method". As a final example, Signatory is now itself a dependency for other libraries (Kidger, 2020).

## 2 BACKGROUND

We begin with some exposition on theory of the signature and logsignature transforms. We begin with definitions and offer intuition afterwards. Also see Reizenstein & Graham (2018) for an introduction focusing on computational concerns, and Lyons et al. (2004) and Hodgkinson et al. (2020) for pedagogical introductions to the motivating theory of rough paths.

## 2.1 THE SIGNATURE TRANSFORM

**Definition 1.** Let $\mathbb{R}^{d_1} \otimes \mathbb{R}^{d_2} \otimes \cdots \otimes \mathbb{R}^{d_n}$ denote the space of all real tensors with shape $d_1 \times d_2 \times \cdots \times d_n$. There is a corresponding binary operation $\otimes$, called the tensor product, which maps a tensor of shape $(d_1, \ldots, d_n)$ and a tensor of shape $(e_1, \ldots, e_m)$ to a tensor of shape $(d_1, \ldots, d_n, e_1, \ldots, e_m)$ via $(A_{i_1, \ldots, i_n}, B_{j_1, \ldots, j_m}) \mapsto A_{i_1, \ldots, i_n} B_{j_1, \ldots, j_m}$. For example when applied to two vectors, it reduces to the outer product.

Let $(\mathbb{R}^d)^{\otimes k} = \mathbb{R}^d \otimes \cdots \otimes \mathbb{R}^d$, and $v^{\otimes k} = v \otimes \cdots \otimes v$ for $v \in \mathbb{R}^d$, in each case with $k-1$ many $\otimes$.

**Definition 2.** Let $N \in \mathbb{N}$. The *signature transform to depth $N$* is defined as

$$\mathrm{Sig}^N \colon \left\{ f \in C([0,1]; \mathbb{R}^d) \;\middle|\; f \text{ differentiable} \right\} \to \prod_{k=1}^{N} \left( \mathbb{R}^d \right)^{\otimes k},$$

$$\mathrm{Sig}^N(f) = \left( \int \cdots \int_{0 < t_1 < \cdots < t_k < 1} \frac{\mathrm{d}f}{\mathrm{d}t}(t_1) \otimes \cdots \otimes \frac{\mathrm{d}f}{\mathrm{d}t}(t_k) \, \mathrm{d}t_1 \cdots \mathrm{d}t_k \right)_{1 \le k \le N}. \tag{1}$$

Most texts define the signature transform using the notation of stochastic calculus. Here, we sacrifice some generality (that is not needed in this context) in favour of more widely-used notation.[2]

The signature transform may naturally be extended to sequences of data.

**Definition 3.** The space of sequences of data over a set $V$ is
$$\mathcal{S}(V) = \{\mathbf{x} = (x_1, \ldots, x_L) \mid L \in \mathbb{N}, \, x_i \in V \text{ for all } i\}.$$
An *interval* of $(x_1, \ldots, x_L) \in \mathcal{S}(V)$ is $(x_i, \ldots, x_j) \in \mathcal{S}(V)$ for some $1 \leq i < j \leq L$.

**Definition 4.** Let $\mathbf{x} = (x_1, \ldots, x_L) \in \mathcal{S}(\mathbb{R}^d)$ with $L \geq 2$. Let $f \colon [0, 1] \to \mathbb{R}^d$ be the unique continuous piecewise affine function such that $f(\frac{i-1}{L-1}) = x_i$ for all $i$, and is affine on the pieces in between. Let $N \in \mathbb{N}$. Then define $\mathrm{Sig}^N(\mathbf{x}) = \mathrm{Sig}^N(f)$. In this way we interpret $\mathrm{Sig}^N$ as a map
$$\mathrm{Sig}^N \colon \mathcal{S}(\mathbb{R}^d) \to \prod_{k=1}^N (\mathbb{R}^d)^{\otimes k}.$$

Note that the choice of $\frac{i-1}{L-1}$ is unimportant; any $L$ points in $[0, 1]$ would suffice, and in fact the definition is invariant to this choice (Bonnier et al., 2019, Definition A.10).

## 2.2 THE GROUPLIKE STRUCTURE

With $A_0 = B_0 = 1 \in \mathbb{R}$ on the right hand side, define $\boxtimes$ by[3]
$$\boxtimes \colon \left(\prod_{k=1}^N (\mathbb{R}^d)^{\otimes k}\right) \times \left(\prod_{k=1}^N (\mathbb{R}^d)^{\otimes k}\right) \to \prod_{k=1}^N (\mathbb{R}^d)^{\otimes k},$$
$$(A_1, \ldots A_N) \boxtimes (B_1, \ldots, B_N) \mapsto \left(\sum_{j=0}^k A_j \otimes B_{k-j}\right)_{1 \leq k \leq N}.$$

Chen's identity (Lyons et al., 2004, Theorem 2.9) states that the image of the signature transform forms a noncommutative group with respect to $\boxtimes$. That is, given a sequence of data $(x_1, \ldots, x_L) \in \mathcal{S}(\mathbb{R}^d)$ and some $j \in \{2, \ldots, L-1\}$, then
$$\mathrm{Sig}^N((x_1, \ldots, x_L)) = \mathrm{Sig}^N((x_1, \ldots, x_j)) \boxtimes \mathrm{Sig}^N((x_j, \ldots, x_L)). \tag{2}$$

Furthermore the signature of a sequence of length two may be computed explicitly from the definition. Letting
$$\exp \colon \mathbb{R}^d \to \prod_{k=1}^N (\mathbb{R}^d)^{\otimes k}, \qquad \exp \colon v \to \left(v, \frac{v^{\otimes 2}}{2!}, \frac{v^{\otimes 3}}{3!}, \ldots, \frac{v^{\otimes N}}{N!}\right),$$
then
$$\mathrm{Sig}^N((x_1, x_2)) = \exp(x_2 - x_1).$$

With Chen's identity, this implies that the signature transform may be computed by evaluating
$$\mathrm{Sig}^N((x_1, \ldots, x_L)) = \exp(x_2 - x_1) \boxtimes \exp(x_3 - x_2) \boxtimes \cdots \boxtimes \exp(x_L - x_{L-1}). \tag{3}$$

## 2.3 THE LOGSIGNATURE, INVERTED SIGNATURE, AND INVERTED LOGSIGNATURE

The group inverse we denote $^{-1}$. Additionally a notion of logarithm may be defined (Liao et al., 2019), where
$$\log \colon \mathrm{image}(\mathrm{Sig}^N) \to \prod_{k=1}^N (\mathbb{R}^d)^{\otimes k}. \tag{4}$$

---

[2]Additionally, many texts also include a $k = 0$ term, which is defined to equal one. We omit this as it does not carry any information, and is therefore irrelevant to the task of machine learning.

[3]Most texts use $\otimes$ rather than $\boxtimes$ to denote this operation, as it may be regarded as an generalisation of the tensor product. That will not be important to us, however, so we use differing notation to aid interpretation.

This then defines the notions of *inverted signature transform*, *logsignature transform* and *inverted logsignature transform* as

$$\text{InvertSig}^N(\mathbf{x}) = \text{Sig}^N(\mathbf{x})^{-1},$$
$$\text{LogSig}^N(\mathbf{x}) = \log\left(\text{Sig}^N(\mathbf{x})\right),$$
$$\text{InvertLogSig}^N(\mathbf{x}) = \log\left(\text{Sig}^N(\mathbf{x})^{-1}\right)$$

respectively. We emphasise that the inverted signature or logsignature transforms are not the inverse maps of the signature or the logsignature transforms.

The logsignature transform extracts the same information as the signature transform, but represents the information in a much more compact way, as $\text{image}\,(\log)$ is a proper subspace[4] of $\prod_{k=1}^{N}\left(\mathbb{R}^d\right)^{\otimes k}$. Its dimension is $w(d, N) = \sum_{k=1}^{N} \frac{1}{k} \sum_{i|k} \mu\left(\frac{k}{i}\right) d^i$, which is known as Witt's formula (Lothaire, 1997). $\mu$ is the Möbius function.

## 2.4 SIGNATURES IN MACHINE LEARNING

In terms of the tensors used by most machine learning frameworks, then the (inverted) signature and logsignature transforms of depth $N$ may both be thought of as consuming a tensor of shape $(b, L, d)$, corresponding to a batch of $b$ different sequences of data, each of the form $(x_1, \ldots, x_L)$ for $x_i \in \mathbb{R}^d$. The (inverted) signature transform then produces a tensor of shape $(b, \sum_{k=1}^{N} d^k)$, whilst the (inverted) logsignature transform produces a tensor of shape $(b, w(d, N))$. We note that these can be easily be large, and much research has focused on ameliorating this Bonnier et al. (2019); Morrill et al. (2020a); Cuchiero et al. (2020).

All of these transforms are in fact differentiable with respect to $\mathbf{x}$, and so may be backpropagated through. These transforms may thus be thought of as differentiable operations between tensors, in the way usually performed by machine learning frameworks.

## 2.5 INTUITION

The (inverted) signature and logsignature transforms all have roughly the same intuition as one another. (They all represent the same information, just in slightly different ways.) Given a sequence of data $(x_1, \ldots, x_L)$, then these transforms may be used as binning functions, feature extractors, or nonlinearities, to give summary statistics over the data.

These summary statistics describe the way in which the data interacts with dynamical systems (Morrill et al., 2020b). Indeed, we have already linked the signature to the exponential map, which is defined as the solution to a differential equation: $\frac{d\exp}{dt}(t) = \exp(t)$. The signature may in fact be defined as the solution of a *controlled* exponential map: $d\text{Sig}^N(f)(t) = \text{Sig}^N(f)(t) \otimes df(t)$, so that $\text{Sig}^N(f)$ is response of a particular dynamical system driven by $f$. The theory here is somewhat involved, and is not an interpretation we shall pursue further here.

An equivalent more straightforward interpretation is arrived at by observing that the terms of the exponential of a scalar $\exp\colon x \in \mathbb{R} \mapsto (1, x, \frac{1}{2}x^2, \ldots)$ produce (up to scaling factors) every monomial of its input. Classical machine learning takes advantage of this as a feature extractor in polynomial regression. The signature transform is the equivalent operation when the input is a sequence.

## 3 CODE EXAMPLE

Signatory is designed to be Pythonic, and offer operations working just like any other PyTorch operation, outputting PyTorch tensors. A brief example is:

```
1  import signatory
2  import torch
3  batch, stream, channels, depth = 1, 10, 2, 4
```

---

[4]log is actually a bijection. $\text{image}\left(\text{Sig}^N\right)$ is some curved manifold in $\prod_{k=1}^{N}\left(\mathbb{R}^d\right)^{\otimes k}$, and log is the map that straightens it out into a linear subspace.

```
4  path = torch.rand(batch, stream, channels, requires_grad=True)
5  signature = signatory.signature(path, depth)
6  signature.sum().backward()
```

## 4 ALGORITHMIC IMPROVEMENTS

We present several noveral algorithmic improvements for computing signatures and logsignatures.

### 4.1 FUSED MULTIPLY-EXPONENTIATE

Recall from equation (3) that the signature may be computed by evaluating several exponentials and several $\boxtimes$. We begin by finding that it is beneficial to compute

$$\left( \prod_{k=1}^{N} \left( \mathbb{R}^d \right)^{\otimes k} \right) \times \mathbb{R}^d \to \prod_{k=1}^{N} \left( \mathbb{R}^d \right)^{\otimes k},$$

$$A, z \mapsto A \boxtimes \exp(z)$$

as a fused operation. Doing so has uniformly (over $d, N$) fewer scalar multiplications than the composition of the individual exponential and $\boxtimes$, and in fact reduces the asymptotic complexity of this operation from $\mathcal{O}(Nd^N)$ to $\mathcal{O}(d^N)$. Furthermore this rate is now optimal, as the size of result (an element of $\prod_{k=1}^{N} \left( \mathbb{R}^d \right)^{\otimes k}$), is itself of size $\mathcal{O}(d^N)$.

The bulk of a signature computation may then be sped up by writing it in terms of this fused operation. See equation (3): a single exponential is required at the start, followed by a reduction with respect to this fused multiply-exponentiate. This gives substantial real-world speedups; see the benchmarks of Section 6.

The fusing is done by expanding

$$A \boxtimes \exp(z) = \left( \sum_{i=0}^{k} A_i \otimes \frac{z^{\otimes(k-i)}}{(k-i)!} \right)_{1 \leq k \leq N},$$

at which point the $k$-th term may be computed by a scheme in the style of Horner's method:

$$\sum_{i=0}^{k} A_i \otimes \frac{z^{\otimes(k-i)}}{(k-i)!} =$$

$$\left( \left( \cdots \left( \left( \frac{z}{k} + A_1 \right) \otimes \frac{z}{k-1} + A_2 \right) \otimes \frac{z}{k-2} + \cdots \right) \otimes \frac{z}{2} + A_{k-1} \right) \otimes z + A_k. \quad (5)$$

See Appendix A.1 for the the mathematics, including proofs of both the asymptotic complexity and the uniformly fewer multiplications.

### 4.2 IMPROVED PRECOMPUTATION STRATEGIES

Given a sequence of data $\mathbf{x} = (x_1, \ldots, x_L)$, it may be desirable to query $\mathrm{Sig}^N((x_i, \ldots, x_j))$ for many different pairs $i, j$. We show that this query may be computed in just $\mathcal{O}(1)$ (in $L$) time and memory by using $\mathcal{O}(L)$ precomputation and storage. Previous theoretical work has achieved only $\mathcal{O}(\log L)$ inference with $\mathcal{O}(L \log L)$ precomputation (Chafai & Lyons, 2005).

Doing so is surprisingly simple. Precompute $\mathrm{Sig}^N((x_1, \ldots, x_j))$ and $\mathrm{InvertSig}^N((x_1, \ldots, x_j))$ for all $j$. This may be done in only $\mathcal{O}(L)$ work, by iteratively computing each signature via

$$\mathrm{Sig}^N((x_1, \ldots, x_j)) = \mathrm{Sig}^N((x_1, \ldots, x_{j-1})) \boxtimes \mathrm{Sig}^N((x_{j-1}, x_j)), \quad (6)$$

with a similar relation for the inverted signature. Then, at inference time use the group-like structure

$$\mathrm{Sig}^N((x_i, \ldots, x_j)) = \mathrm{InvertSig}^N((x_1, \ldots, x_i)) \boxtimes \mathrm{Sig}^N((x_1, \ldots, x_j)),$$

followed by a $\log$ if it is a logsignature that is desired. As a single operation this is $\mathcal{O}(1)$ in $L$.

We do remark that this should be used with caution, and may suffer from numerical stability issues when used for large $i, j$.

### 4.3 MORE EFFICIENT LOGSIGNATURE BASIS

The logsignature transform of a path has multiple possible representations, corresponding to different possible bases of the ambient space, which is typically interpreted as a free Lie algebra (Reutenauer, 1993). The *Lyndon basis* is a typical choice (Reizenstein & Graham, 2018).

We show that there exists a more computationally efficient basis. It is mathematically unusual, as it is not constructed as a Hall basis. But if doing deep learning, then the choice of basis is (mostly) unimportant if the next operation is a learnt linear transformation.

The Lyndon basis uses *Lyndon brackets* as its basis elements. Meanwhile our new basis uses basis elements that, when written as a sum of Lyndon brackets and expanded as a sum of words, have precisely one word as a Lyndon word. This means that the coefficient of this basis element can be found cheaply, by extracting the coeffient of that Lyndon word from the tensor algebra representation of the logsignature. See Appendix A.2 for the full exposition.

## 5 NEW FEATURES

Signatory provides several features not available in previous libraries.

### 5.1 PARALLELISM

There are two main levels of parallelism. First is naïve parallelism over the batch dimension. Second, we observe that equation (3) takes the form of a noncommutative reduction with respect to the fused multiply-exponentiate. The operation $\boxtimes$ is associative, and so this may be parallelised in the usual way for reductions, by splitting the computation up into chunks.

Parallelism on the CPU is implemented with OpenMP. For speed the necessary operations are written in terms of C++ primitives, and then bound into PyTorch.

### 5.2 GPU SUPPORT

An important feature is GPU support, which is done via the functionality available through LibTorch. It was a deliberate choice not to write CUDA code; this is due to technical reasons for distributing the library in a widely-compatible manner. See Appendix B.

There are again two levels of parallelism, as described in the previous section.

When the (log)signature transform is part of a deep learning model trained on the GPU, then GPU support offers speedups not just from the use of a GPU over the CPU, but also obviates the need for copying the data to and from the GPU.

### 5.3 BACKPROPAGATION

Crucial for any library used in deep learning is to be able to backpropagate through the provided operations. Signatory provides full support for backpropagation through every provided operation. There has previously only been limited support for backpropagation through a handful of simple operations, via the `iisignature` library (Reizenstein & Graham, 2018).

The backpropagation computations are handwritten, rather than being generated autodifferentiably. This improves the speed of the computation by using C++ primitives rather than high-level tensors, and furthermore allows for improved memory efficiency, by exploiting a reversibility property of the signature (Reizenstein, 2019, Section 4.9.3). We discuss backpropagation in more detail in Appendix C.

### 5.4 INVERTED SIGNATURES AND LOGSIGNATURES

Signatory provides the capability to compute inverted signatures and logsignatures, via the optional `inverse` argument to the `signature` and `logsignature` functions. This is primarily a convenience as

$$\mathrm{Sig}^N((x_1, \ldots, x_L))^{-1} = \mathrm{Sig}^N((x_L, \ldots, x_1)).$$

## 5.5 Exploiting the grouplike structure

It is often desirable to compute (inverted) (log)signatures over multiple intervals of the same sequence of data. These calculations may jointly be accomplished more efficiently than by evaluating the signature transform over every interval separately. In some cases, if the original data has been discarded and only its signature is now known, exploiting this structure is the only way to perform the computation.

Here we detail several notable cases, and how Signatory supports them. In all cases the aim is to provide a flexible set of tools that may be used together, so that wherever possible unecessary recomputation may be elided. Their use is also discussed in the documentation, including examples.

**Combining adjacent intervals**    Recall equation (2). If the two signatures on the right hand side of the equation are already known, then the signature of the overall sequence of data may be computed using only a single $\boxtimes$ operation, without re-iterating over the data.

This operation is provided for by the `multi_signature_combine` and `signature_combine` functions.

**Expanding intervals**    Given a sequence of data $(x_1, \ldots, x_L) \in \mathcal{S}\left(\mathbb{R}^d\right)$, a scenario that is particularly important for its use in Section 4.2 is to compute the signature of expanding intervals of the data,[5]

$$(\mathrm{Sig}^N((x_1, x_2)), \mathrm{Sig}^N((x_1, x_2, x_3)), \ldots, \mathrm{Sig}^N((x_1, \ldots, x_L))).$$

This may be interpreted as a sequence of signatures, that is to say an element of $\mathcal{S}\left(\prod_{k=1}^{N}\left(\mathbb{R}^d\right)^{\otimes k}\right)$.

By equation (6), this may be done efficiently in only $\mathcal{O}(L)$ work, and with all the earlier signatures available as byproducts, for free, of computing the final element $\mathrm{Sig}^N((x_1, \ldots, x_L))$.

This is handled by the optional `stream` argument to the `signature` function and to the `logsignature` function.

**Arbitrary intervals**    Given a sequence of data $\mathbf{x} = (x_1, \ldots, x_L)$, it may be desirable to query $\mathrm{Sig}^N((x_i, \ldots, x_j))$ for many $i, j$ such that $1 \leq i < j \leq L$, as in Section 4.2. Using the efficient precomputation strategy described there, Signatory provides this capability via the `Path` class.

**Keeping the signature up-to-date**    Suppose we have a sequence of data $(x_1, \ldots, x_L) \in \mathcal{S}\left(\mathbb{R}^d\right)$ whose signature $\mathrm{Sig}^N((x_1, \ldots, x_L))$ has already been computed. New data subsequently arrives, some $(x_{L+1}, \ldots, x_{L+M}) \in \mathcal{S}\left(\mathbb{R}^d\right)$, and we now wish to update our computed signature, for example to compute the sequence of signatures

$$(\mathrm{Sig}^N((x_1, \ldots, x_{L+1})), \ldots, \mathrm{Sig}^N((x_1, \ldots, x_{L+M}))). \tag{7}$$

This could done by computing the signatures over $(x_{L+1}, \ldots, x_{L+M})$, and combining them together as above. However, if these signatures (over $(x_{L+1}, \ldots, x_{L+M})$) are not themselves of interest, then this approach may be improved upon, as this only exploits the grouplike structure, but not the fused multiply-exponentiate described in Section 4.1.

Computing them in this more efficient way may be handled via the `basepoint` and `initial` arguments to the `signature` function, and via the `update` method of the `Path` class.

## 6 Benchmark performance

We are aware of two existing software libraries providing similar functionality, `esig` (Lyons, 2017) and `iisignature` (Reizenstein & Graham, 2018). We ran a series of benchmarks against the latest versions of both of these libraries, namely `esig` 0.6.31 and `iisignature` 0.24. The computer used was equipped with a Xeon E5-2960 v4 and a Quadro GP100, and was running Ubuntu 18.04 and Python 3.7.

---

[5]Note that we start with $\mathrm{Sig}^N((x_1, x_2))$, as two is the shortest a sequence of data can be to define a path; see Definition 4.

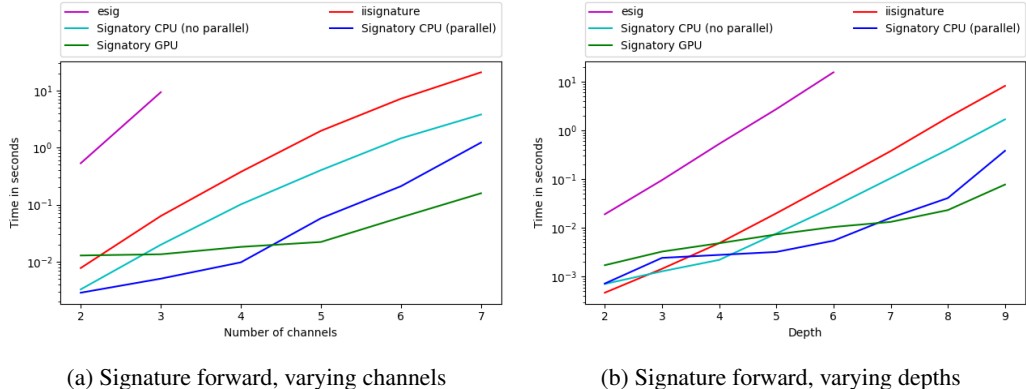

(a) Signature forward, varying channels        (b) Signature forward, varying depths

Figure 1: Time taken on benchmark computations to compute the signature transform. `esig` is only shown for small operations as it is incapable of larger operations. Note the logarithmic scale.

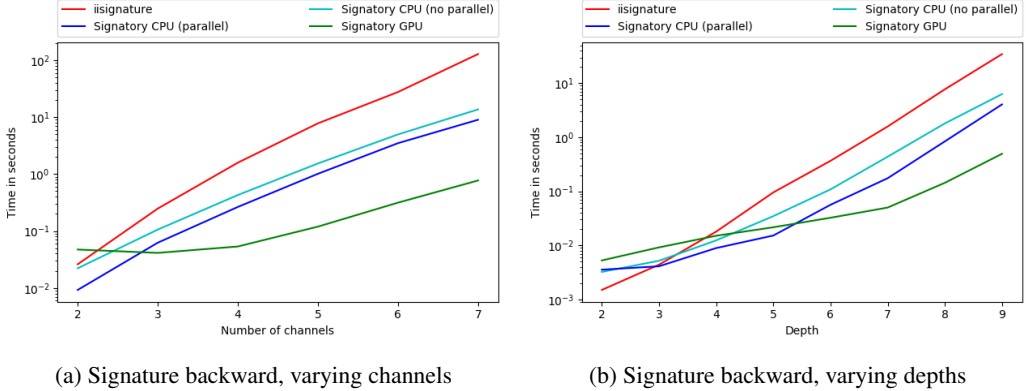

(a) Signature backward, varying channels       (b) Signature backward, varying depths

Figure 2: Time taken on benchmark computations to backpropagate through the signature transform. `esig` is not shown as it is incapable of computing backward operations. Note the logarithmic scale.

## 6.1 DIRECT COMPARISON

For `esig` and `iisignature` we report run time on the CPU, whilst for Signatory we report run time on the GPU, CPU with parallelism, and CPU without parallelism. As it is in principle possible to parallelise these alternative libraries using Python's `multiprocessing` module[6], the most important comparisons are to Signatory on the CPU without parallelism (representing like-for-like computational resources), and on the GPU (representing the best possible performance).

We begin with a benchmark for the forward operation through the signature transform. We consider a batch of 32 sequences, of length 128. We then investigate the scaling as we vary either the number of channels (over 2–7) in the input sequences, or the depth (over 2–9) of the signature transform. For varying channels, the depth was fixed at 7. For varying depths, the channels was fixed at 4.

Every test case is repeated 50 times and the fastest time taken. See Figure 1. Note the logarithmic scale.

We observe that `iisignature` is Signatory's strongest competitor in all cases. Signatory and `iisignature` are comparable for the very smallest of computations. As the computation increases in size, then the CPU implementations of Signatory immediately overtake `iisignature`, followed by the GPU implementation.

For larger computations, Signatory can be orders of magnitude faster. For example, to compute the signature transform with depth and number of channels both equal to 7, then `iisignature` takes 20.9 seconds to perform the computation. In contrast, running on the CPU without parallelism,

---

[6]Subject to nontrivial overhead.

Signatory takes only 3.8 seconds, which represents a 5.5× speedup. We emphasise that the same computational resources (including lack of parallelism) were used for both. To see the benefits of a GPU implementation over a CPU implementation—the primary motivation for Signatory's existence—then we observe that Signatory takes only 0.16 seconds to compute this same operation. Compared to the best previous alternative in `iisignature`, this represents a 132× speedup.

Next, we consider the backward operation through the signature transform. We vary over multiple inputs as before. See Figure 2. We again observe the same behaviour. `iisignature` is Signatory's strongest competitor, but is still orders of magnitude slower on anything but the very smallest of problems. For example, to backpropagate through the signature transform, with depth and number of channels both equal to 7, then Signatory on the CPU without parallelism takes 13.7 seconds. Meanwhile, `iisignature` takes over 2 minutes – 128 seconds – to perform this computation on like-for-like computational resources. Running Signatory on the GPU takes a fraction of a second, specifically 0.772 seconds. These represent speedups of 9.4× and 166× respectively.

For further speed benchmarks, a discussion on memory usage benchmarks, the precise numerical values of the graphs presented here, and code to reproduce these benchmarks, see Appendix D. We observe the same consistent improvements on these additional benchmarks.

## 6.2 DEEP LEARNING EXAMPLE

To emphasise the benefit of Signatory to deep learning applications, we consider training a deep signature model (Bonnier et al., 2019) on a toy dataset of geometric Brownian motion samples. The samples have one of two different volatilities, and the task is to perform binary classification.

The model sweeps a small feedforward network over the input sequence (to produce a sequence of hidden states), applies the signature transform, and then maps to a binary prediction via a final learnt linear map. This model has learnt parameters prior to the signature transform, and so in particular backpropagation through the signature transform is necessary.

The signature transform is computed either using Signatory, or using `iisignature`. We train the model on the GPU and plot training loss against wall-clock time.

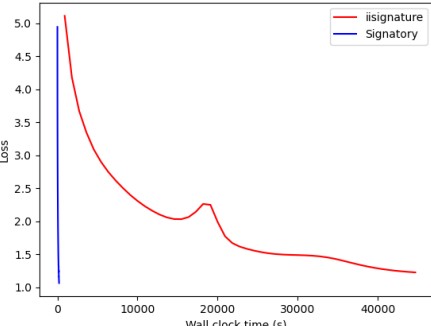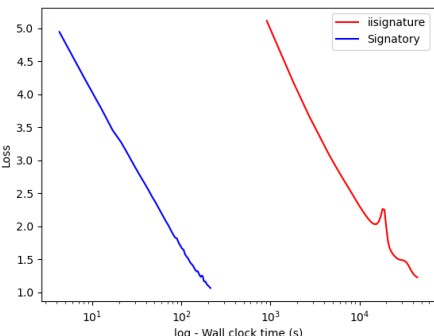

Figure 3: Loss against wall-clock time for a deep signature model. Both plots are identical; the second plot uses a log-scaled x-axis.

Both models train successfully, but the model using Signatory trains 210 times faster than the one using `iisignature`. This makes clear how signatures have previously represented the largest computational bottleneck. The improvement of 210× is even larger than the improvements obtained in the previous section. We attribute this to the fact that `iisignature` necessarily has the additional overhead copying data from the GPU to the CPU and back again.

## 7 CONCLUSION

We have introduced Signatory, a library for performing functionality related to the signature and logsignature transforms, with a particular focus on applications to machine learning. Notable contributions are the speed of its operation, its GPU support, differentiability of every provided operation, and its novel algorithmic innovations.

ACKNOWLEDGEMENTS

This work was supported by the Engineering and Physical Sciences Research Council [EP/L015811/1].

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

## A    FURTHER DETAILS OF ALGORITHMIC IMPROVEMENTS

### A.1    FUSED MULTIPLY-EXPONENTIATE

The conventional way to compute a signature is to iterate through the computation described by equation (3): for each new increment, take its exponential, and $\boxtimes$ it on to what has already been computed; repeat.

Our proposed alternate way is to fuse the exponential and $\boxtimes$ into a single operation, and then iteratively perform this fused operation.

We now count the number of multiplications required to compute

$$\left( \prod_{k=1}^{N} \left(\mathbb{R}^d\right)^{\otimes k} \right) \times \mathbb{R}^d \rightarrow \prod_{k=1}^{N} \left(\mathbb{R}^d\right)^{\otimes k},$$

$$A, z \mapsto A \boxtimes \exp(z)$$

for each approach.

We will establish that the fused operation uses fewer multiplications for all possible $d \geq 1$ and $N \geq 1$. We will then demonstrate that it is in fact of a lower asymptotic complexity.

### A.1.1    THE CONVENTIONAL WAY

The exponential is defined as

$$\exp\colon \mathbb{R}^d \rightarrow \prod_{k=1}^{N} \left(\mathbb{R}^d\right)^{\otimes k},$$

$$\exp\colon x \mapsto \left( x, \frac{x^{\otimes 2}}{2!}, \frac{x^{\otimes 3}}{3!}, \ldots, \frac{x^{\otimes N}}{N!} \right),$$

see Bonnier et al. (2019, Proposition 15).

Note that every tensor in the exponential is symmetric, and so in principle requires less work to compute than its number of elements would suggest. For the purposes of this analysis, to give the benefit of the doubt to a competing method, we shall assume that this is done (although taking advantage of this in practice is actually quite hard (Reizenstein & Graham, 2018, Section 2)). This takes

$$\sum_{k=2}^{N} \left( d + \binom{d+k-1}{k} \right)$$

scalar multiplications, using the formula for unordered sampling with replacement (Reizenstein & Graham, 2018, Section 2), under the assumption that each division by a scalar costs the same as a multiplication (which can be accomplished by precomputing their reciprocals and then multiplying by them).

Next, we need to count the number of multiplications to perform a single $\boxtimes$.

Let

$$A, B \in \prod_{k=1}^{N} \left(\mathbb{R}^d\right)^{\otimes k}.$$

Let $A = (A_1, \ldots, A_N)$, with

$$A_i = (A_i^{j_1, \ldots, j_i})_{1 \leq j_1, \ldots, j_i \leq d},$$

and every $A_i^{j_1, \ldots, j_i} \in \mathbb{R}$. Additionally let $A_0 = 1$. Similarly for $B$. Then $\boxtimes$ is defined by

$$\boxtimes\colon \left( \prod_{k=1}^{N} \left(\mathbb{R}^d\right)^{\otimes k} \right) \times \left( \prod_{k=1}^{N} \left(\mathbb{R}^d\right)^{\otimes k} \right) \rightarrow \prod_{k=1}^{N} \left(\mathbb{R}^d\right)^{\otimes k},$$

$$\boxtimes\colon A, B \mapsto \left( \sum_{i=0}^{k} A_i \otimes B_{k-i} \right)_{1 \leq k \leq N}, \tag{8}$$

where each

$$A_i \otimes B_{k-i} = \left( A_i^{j_1,\ldots,j_i} B_{k-i}^{\hat{j}_1,\ldots,\hat{j}_{k-i}} \right)_{1 \leq j_1,\ldots,j_i,\hat{j}_1,\ldots,\hat{j}_{k-i} \leq d}$$

is the usual tensor product, the result is thought of as a tensor in $(\mathbb{R}^d)^{\otimes k}$, and the summation is taken in this space. See Bonnier et al. (2019, Definition A.13).

To the authors' knowledge there has been no formal analysis of a lower bound on the computational complexity of $\boxtimes$, and there is no better way to compute it than naïvely following this definition.

This, then, requires

$$\sum_{k=1}^{N} \sum_{i=1}^{k-1} \sum_{j_1,\ldots,j_i=1}^{d} \sum_{\hat{j}_1,\ldots,\hat{j}_{k-i}=1}^{d} 1 = \sum_{k=1}^{N} \sum_{i=1}^{k-1} d^k$$

$$= \sum_{k=1}^{N} (k-1)d^k$$

scalar multiplications.

Thus the overall cost of the conventional way involves

$$\mathcal{C}(d, N) = \sum_{k=2}^{N} \left( d + \binom{d+k-1}{k} \right) + \sum_{k=1}^{N} (k-1)d^k \tag{9}$$

scalar multiplications.

### A.1.2 THE FUSED OPERATION

Let $A \in \prod_{k=1}^{N} \left( \mathbb{R}^d \right)^{\otimes k}$ and $z \in \mathbb{R}^d$. Then

$$A \boxtimes \exp(z) = \left( \sum_{i=0}^{k} A_i \otimes \frac{z^{\otimes(k-i)}}{(k-i)!} \right)_{1 \leq k \leq N},$$

where the $k$-th term may be computed by a scheme in the style of Horner's method:

$$\sum_{i=0}^{k} A_i \otimes \frac{z^{\otimes(k-i)}}{(k-i)!} =$$
$$\left( \left( \cdots \left( \left( \frac{z}{k} + A_1 \right) \otimes \frac{z}{k-1} + A_2 \right) \otimes \frac{z}{k-2} + \cdots \right) \otimes \frac{z}{2} + A_{k-1} \right) \otimes z + A_k. \tag{10}$$

As before, we assume that the reciprocals $\frac{1}{2}, \ldots, \frac{1}{N}$ have been precomputed, so that each division costs the same as a multiplication.

Then we begin by computing $z/2, \ldots, z/N$, which takes $d(N-1)$ multiplications.

Computing the $k$-th term as in equation (10) then involves $d^2 + d^3 + \cdots + d^k$ multiplications. This is because, working from innermost bracket to outermost, the first $\otimes$ produces a $d \times d$ matrix as the outer product of two size $d$ vectors, and may thus be computed with $d^2$ multiplications; the second $\otimes$ produces a $d \times d \times d$ tensor from a $d \times d$ matrix and a size $d$ vector, and may thus be computed with $d^3$ multiplications; and so on.

Thus the overall cost of a fused multiply-exponentiate is

$$\mathcal{F}(d, N) = d(N-1) + \sum_{k=1}^{N} \sum_{i=2}^{k} d^i \tag{11}$$

scalar multiplications.

### A.1.3 COMPARISON

We begin by establishing the uniform bound $\mathcal{F}(d, N) \leq \mathcal{C}(d, N)$ for all $d \geq 1$ and $N \geq 1$.

First suppose $d = 1$. Then

$$\mathcal{F}(1, N) = (N - 1) + \sum_{k=1}^{N}(k - 1)$$

$$\leq 2(N - 1) + \sum_{k=1}^{N}(k - 1)$$

$$= \mathcal{C}(1, N).$$

Now suppose $N = 1$. Then

$$\mathcal{F}(d, 1) = 0 = \mathcal{C}(d, 1).$$

Now suppose $N = 2$. Then

$$\mathcal{F}(d, 2) = d + d^2$$

$$\leq d + \binom{d + 1}{2} + d^2$$

$$= \mathcal{C}(d, 2)$$

Now suppose $d \geq 2$ and $N \geq 3$. Then

$$\mathcal{F}(d, N) = d(N - 1) + \sum_{k=1}^{N}\sum_{i=2}^{k} d^i$$

$$= \frac{d^{N+2} - d^3 - (N - 1)d^2 + (N - 1)d}{(d - 1)^2}. \tag{12}$$

And

$$\mathcal{C}(d, N) = \sum_{k=2}^{N}\left(d + \binom{d + k - 1}{k}\right) + \sum_{k=1}^{N}(k - 1)d^k$$

$$\geq \sum_{k=1}^{N}(k - 1)d^k$$

$$= \frac{(N - 1)d^{N+2} - Nd^{N+1} + d^2}{(d - 1)^2}. \tag{13}$$

Thus we see that it suffices to show that

$$d^{N+2} - d^3 - (N - 1)d^2 + (N - 1)d \leq (N - 1)d^{N+2} - Nd^{N+1} + d^2,$$

for $d \geq 2$ and $N \geq 3$. That is,

$$0 \leq d^{N+1}(d(N - 2) - N) + d(d^2 + N(d^2 - 1) + 1). \tag{14}$$

At this point $d = 2$, $N = 3$ must be handled as a special case, and may be verified by direct evaluation of equation (14). So now assume $d \geq 2$, $N \geq 3$, and that $d = 2$, $N = 3$ does not occur jointly. Then we see that equation (14) is implied by

$$0 \leq d(N - 2) - N \quad \text{and} \quad 0 \leq d^2 + N(d^2 - 1) + 1.$$

The second condition is trivially true. The first condition rearranges to $N/(N - 2) \leq d$, which is now true for $d \geq 2$, $N \geq 3$ with $d = 2$, $N = 3$ not jointly true.

This establishes the uniform bound $\mathcal{F}(d, N) \leq \mathcal{C}(d, N)$.

Checking the asymptotic complexity is much more straightforward. Consulting equations (12) and (13) shows that $\mathcal{F}(d, n) = \mathcal{O}(d^N)$ whilst $\mathcal{C}(d, N) = \Omega(Nd^N)$. And in fact as $\binom{d+k-1}{k} \leq d^k$ then equation (9) demonstrates that $\mathcal{C}(d, N) = \mathcal{O}(Nd^N)$.

## A.2 LOGSIGNATURE BASES

We move on to describing our new more efficient basis for the logsignature.

### A.2.1 WORDS, LYNDON WORDS, AND LYNDON BRACKETS

Let $\mathcal{A} = \{a_1, \ldots, a_d\}$ be a set of $d$ letters. Let $\mathcal{A}^{+N}$ be the set of all words in these letters, of length between 1 and $N$ inclusive. For example $a_1 a_4 \in \mathcal{A}^{+N}$ is a word of length two.

Impose the order $a_1 < a_2 < \cdots < a_d$ on $\mathcal{A}$, and extend it to the lexicographic order on words in $\mathcal{A}^{+N}$ of the same length as each other, so that for example $a_1 a_2 < a_1 a_3 < a_2 a_1$. Then a *Lyndon word* (Lalonde & Ram, 1995) is a word which comes earlier in lexicographic order than any of its rotations, where rotation corresponds to moving some number of letters from the start of the word to the end of the word. For example $a_2 a_2 a_3 a_4$ is a Lyndon word, as it is lexicographically earlier than $a_2 a_3 a_4 a_2$, $a_3 a_4 a_2 a_2$ and $a_4 a_2 a_2 a_3$. Meanwhile $a_2 a_2$ is not a Lyndon word, as it is not lexicographically earlier than $a_2 a_2$ (which is a rotation). Denote by $\mathcal{L}\left(\mathcal{A}^{+N}\right)$ the set of all Lyndon words of length between 1 and $N$.

Given any Lyndon word $w_1 \cdots w_n$ with $n \geq 2$ and $w_i \in \mathcal{A}$, we may consider its *longest Lyndon suffix*; that is, the smallest $j > 1$ for which $w_j \cdots w_n$ is a Lyndon word. (It is guaranteed to exist as $w_n$ alone is a Lyndon word.) It is a fact (Lalonde & Ram, 1995) that $w_1 \cdots w_{j-1}$ is then also a Lyndon word. Given a Lyndon word $w$, we denote by $w^b$ its longest Lyndon suffix, and by $w^a$ the corresponding prefix.

Considering spans with respect to $\mathbb{R}$, let

$$[\cdot, \cdot] \colon \operatorname{span}(\mathcal{A}^{+N}) \times \operatorname{span}(\mathcal{A}^{+N}) \to \operatorname{span}(\mathcal{A}^{+N})$$

be the commutator given by

$$[w, z] = wz - zw,$$

where $wz$ denotes concatenation of words, distributed over the addition, as $w$ and $z$ belong to a span and thus may be linear combinations of words. For example $w = 2a_1 a_2 + a_1$ and $z = a_1 + a_3$ gives $wz = 2a_1 a_2 a_1 + 2a_1 a_2 a_3 + a_1 a_1 + a_1 a_3$.

Then define

$$\phi \colon \mathcal{L}\left(\mathcal{A}^{+N}\right) \to \operatorname{span}(\mathcal{A}^{+N})$$

by $\phi(w) = w$ if $w$ is a word of only a single letter, and by

$$\phi(w) = [\phi(w^a), \phi(w^b)]$$

otherwise. For example,

$$\begin{aligned}
\phi(a_1 a_2 a_2) &= [[a_1, a_2], a_2] \\
&= [a_1 a_2 - a_2 a_1, a_2] \\
&= a_1 a_2 a_2 - 2a_2 a_1 a_2 + a_2 a_2 a_1.
\end{aligned}$$

Now extend $\phi$ by linearity from $\mathcal{L}\left(\mathcal{A}^{+N}\right)$ to $\operatorname{span}(\mathcal{L}\left(\mathcal{A}^{+N}\right))$, so that

$$\phi \colon \operatorname{span}(\mathcal{L}\left(\mathcal{A}^{+N}\right)) \to \operatorname{span}(\mathcal{A}^{+N})$$

is a linear map between finite dimensional real vector spaces, from a lower dimensional space to a higher dimensional space.

Next, let

$$\psi \colon \mathcal{A}^{+N} \to \operatorname{span}(\mathcal{L}\left(\mathcal{A}^{+N}\right))$$

be such that $\psi(w) = w$ if $w \in \mathcal{L}\left(\mathcal{A}^{+N}\right)$, and $\psi(w) = 0$ otherwise. Extend $\psi$ by linearity to $\operatorname{span}(\mathcal{A}^{+N})$, so that

$$\psi \colon \operatorname{span}(\mathcal{A}^{+N}) \to \operatorname{span}(\mathcal{L}\left(\mathcal{A}^{+N}\right))$$

is a linear map between finite dimensional real vector spaces, from a higher dimensional space to a lower dimensional space.

### A.2.2   A BASIS FOR SIGNATURES

Recall that the signature transform maps between spaces as follows.

$$\mathrm{Sig}^N \colon \mathcal{S}\left(\mathbb{R}^d\right) \to \prod_{k=1}^{N} \left(\mathbb{R}^d\right)^{\otimes k}.$$

Let $\{e_i \mid 1 \le i \le d\}$ be the usual basis for $\mathbb{R}^d$. Then

$$\{e_{i_1} \otimes \cdots \otimes e_{i_k} \mid 1 \le i_1, \ldots i_k \le d\}$$

is a basis for $(\mathbb{R}^d)^{\otimes k}$. An arbitrary element of $\prod_{k=1}^{N} \left(\mathbb{R}^d\right)^{\otimes k}$ may be written as

$$\left( \sum_{i_1, \ldots i_k = 1}^{d} \alpha_{i_1, \ldots, i_k} e_{i_1} \otimes \cdots \otimes e_{i_k} \right)_{1 \le k \le N} \tag{15}$$

for some $\alpha_{i_1, \ldots, i_k}$.

Then $\mathcal{A}^{+N}$ may be used to represent a basis for $\prod_{k=1}^{N} \left(\mathbb{R}^d\right)^{\otimes k}$. Identify $e_{i_1} \otimes \cdots \otimes e_{i_k}$ with $a_{i_1} \cdots a_{i_k}$. Extend linearly, so as to identify expression (15) with the formal sum of words

$$\sum_{k=1}^{N} \sum_{i_1, \ldots i_k = 1}^{d} \alpha_{i_1, \ldots, i_k} a_{i_1} \cdots a_{i_k}.$$

With this identification,

$$\mathrm{span}(\mathcal{A}^{+N}) \cong \prod_{k=1}^{N} \left(\mathbb{R}^d\right)^{\otimes k} \tag{16}$$

### A.2.3   BASES FOR LOGSIGNATURES

Suppose we have some $\mathbf{x} \in \mathcal{S}\left(\mathbb{R}^d\right)$. Using the identification in equation (16), then we may attempt to seek some $x \in \mathrm{span}(\mathcal{L}\left(\mathcal{A}^{+N}\right))$ such that

$$\phi(x) = \log\left(\mathrm{Sig}^N(\mathbf{x})\right). \tag{17}$$

This is an overdetermined linear system. As a matrix $\phi$ is tall and thin. However it turns out that $\mathrm{image}\,(\log) = \mathrm{image}\,(\phi)$ and moreover there exists a unique solution (Reizenstein & Graham, 2018). (That it is an overdetermined system is typically the point of the logsignature transform over the signature transform, as it then represents the same information in less space.)

If $x = \sum_{\ell \in \mathcal{L}(\mathcal{A}^{+N})} \alpha_\ell \ell$, with $\alpha_\ell \in \mathbb{R}$, then by linearity

$$\sum_{\ell \in \mathcal{L}(\mathcal{A}^{+N})} \alpha_\ell \phi(\ell) = \log\left(\mathrm{Sig}^N(\mathbf{x})\right),$$

so that $\phi(\mathcal{L}\left(\mathcal{A}^{+N}\right))$ is a basis, called the Lyndon basis, of $\mathrm{image}\,(\log)$. When calculating the logsignature transform in a computer, then the collection of $\alpha_\ell$ are a sensible choice for representing the result, and indeed, this is what is done by `iisignature`. See Reizenstein & Graham (2018) for details of this procedure.

However, it turns out that this is unnecessarily expensive. In deep learning, it is typical to apply a learnt linear transformation after a nonlinearity - in which case we largely do not care in what basis we represent the logsignature, and it turns out that we can find a more efficient one.

The Lyndon basis exhibits a particular triangularity property (Reutenauer, 1993, Theorem 5.1), (Reizenstein, 2019, Theorem 32), meaning that for all $\ell \in \mathcal{L}\left(\mathcal{A}^{+N}\right)$, then $\phi(\ell)$ has coefficient zero for any Lyndon word lexicographically earlier than $\ell$. This property has already been exploited by `iisignature` to solve (17) efficiently, but we can do better: it means that

$$\psi \circ \phi \colon \mathrm{span}\left(\mathcal{L}\left(\mathcal{A}^{+N}\right)\right) \to \mathrm{span}\left(\mathcal{L}\left(\mathcal{A}^{+N}\right)\right)$$

is a triangular linear map, and so in particular it is invertible, and defines a change of basis; it is this alternate basis that we shall use instead. Instead of seeking $x$ as in equation (17), we may now instead seek $z \in \mathrm{span}\left(\mathcal{L}\left(\mathcal{A}^{+N}\right)\right)$ such that

$$(\phi \circ (\psi \circ \phi)^{-1})(z) = \log\left(\mathrm{Sig}^N(\mathbf{x})\right).$$

But now by simply applying $\psi$ to both sides:

$$z = \psi\left(\log\left(\mathrm{Sig}^N(\mathbf{x})\right)\right).$$

This is now incredibly easy to compute. Once $\log\left(\mathrm{Sig}^N(\mathbf{x})\right)$ has been computed, and interpreted as in equation (16), then the operation of $\psi$ is simply to extract the coefficients of all the Lyndon words, and we are done.

## B  LIBTORCH VS CUDA

LibTorch is the C++ equivalent to the PyTorch library. GPU support in Signatory was provided by using the operations provided by LibTorch.

It was a deliberate choice not to write custom CUDA kernels. The reason for this is as follows. We have to make a choice between distributing source code and distributing precompiled binaries. If we distribute source code, then we rely on users being able to compile CUDA, which is far from a guarantee.

Meanwhile, distributing precompiled binaries is unfortunately not feasible on Linux. C/C++ extensions for Python are typically compiled for Linux using the 'manylinux' specification, and indeed PyPI will only host binaries claiming to be compiled according to this specification. Unfortunately, based on our inspection of its build scripts, PyTorch appears not to conform to this specification. It instead compiles against a later version of Centos than is supported by manylinux, and then subsequently modifies things so as to *seem* compatible with the manylinux specification.

Unpicking precisely how PyTorch does this so that we might duplicate the necessary functionality (as we must necessarily remain compatible with PyTorch as well) was judged a finickity task full of hard-to-test edge cases, that is an implementation detail of PyTorch that should not be relied upon, and that may not remain stable across future versions.

## C  BACKPROPAGATION

Backpropagation is calculated in the usual way, mathematically speaking, by treating the signature and logsignature transforms as a composition of differentiable primitives, as discussed in Section 2.2.

The backpropagation computations are handwritten, and are not generated autodifferentiably. This improves the speed of the computation by using C++ primitives, rather than high-level tensors.

### C.1  REVERSIBILITY

Moreover, it allows us to exploit a reversibility property of the signature (Reizenstein, 2019). When backpropagating through any forward operation, then typically the the forward results are stored in memory, as these are used in the backward pass.

However, recall the grouplike structure of the signature; in particular this means that

$$\mathrm{Sig}^N((x_1, \ldots, x_{L-1})) = \mathrm{Sig}^N((x_1, \ldots, x_L)) \boxtimes \mathrm{Sig}^N((x_{L-1}, x_L))^{-1}.$$
$$= \mathrm{Sig}^N((x_1, \ldots, x_L)) \boxtimes \mathrm{Sig}^N((x_L, x_{L-1})). \tag{18}$$

Consider the case of $\mathrm{Sig}^N((x_1, \ldots, x_L))$ by iterating through equation (3) from left to right. Reversibility means we do not need to store the intermediate computations $\mathrm{Sig}^N((x_1, \ldots, x_j))$: given the final $\mathrm{Sig}^N((x_1, \ldots, x_L))$, we can recover $\mathrm{Sig}^N((x_1, \ldots, x_j))$ in the order that they are needed in the backward pass by repeatedly applying equation (18).

We remark in Section 2.5 that the signature may be interpreted as the solution to a differential equation. This recomputation procedure actually corresponds to the adjoint method for backpropagating through a differential equation, as popularised in machine learning via Chen et al. (2018).

Importantly however, this does not face reconstruction errors in the same way as neural differential equations (Gholami et al., 2019). Because the driving path $f$ is taken to be piecewise affine in Definition 4, then the differential equation defining the signature may be solved exactly, without numerical approximations.

Equation (18) uses the same basic operations as the forward operation, and can be computed using the same subroutines, including the fused multiply-exponentiate.

### C.2 SPEED VERSUS MEMORY TRADE-OFFS

The reversibility procedure just described introduces the additional cost of recomputing the path (rather than just holding it in memory). In principle this need not be performed by holding partial results in memory.

For simplicity we do not offer this an alternative with Signatory. Signature-based techniques are often applied to long or high-frequency data (Lyons et al., 2014; Morrill et al., 2020b), for which the large size of multiple partially computed signatures can easily become a memory issue. Nonetheless this represents an opportunity for further work.

### C.3 PARALLELISM

The use of parallelism in the gradient computation depends upon whether to use reversibility as discussed.

Consider first the case in which reversibility is not used, and all intermediate results are held in memory. As discussed in Section 5.1, the forward operation may be computed in parallel as a reduction. The computation graph (within the signature computation) then looks like a balanced tree, and so the backward operation through this computation graph may be performed in parallel as well.

However if reversibility is used then only the final $\mathrm{Sig}^N((x_1, \ldots, x_L))$ is held in memory, then the necessary intermediate computations to backpropagate in parallel are not available.

As Signatory uses reversibility then backpropagation is not performed in parallel.

This represents an opportunity for further work, but practically speaking we expect that its impact is only moderate. Backpropagation is typically performed as part of a training procedure over batches of data; thus the available parallelism may already saturated by parallelism over the batch, and by the intrinsic parallelism available within each primitive operation.

## D  FURTHER BENCHMARKS

### D.1  CODE FOR REPRODUCIBILITY

The benchmarks may be reproduced with the following code on a Linux system. First we install the necessary packages.

```
pip install numpy==1.18.0 matplotlib==3.0.3 torch==1.5.0
pip install iisignature==0.24 esig==0.6.31 signatory==1.2.1.1.5.0
git clone https://github.com/patrick-kidger/signatory.git
cd signatory
```

Note that numpy must be installed before `iisignature`, and PyTorch must be installed before Signatory. The unusually long version number for Signatory is necessary to specify both the version of Signatory, and the version of PyTorch that it is for. The `git clone` is necessary as the benchmarking code is not distributed via `pip`.

Now run

```
python command.py benchmark -help
```

for further details on how to run any particular benchmark. For example,

```
python command.py benchmark -m time -f sigf -t channels -o graph
```

will reproduce Figure 1a.

## D.2 MEMORY BENCHMARKS

Our benchmark scripts offer some limited ability to benchmark memory consumption, via the `-m memory` flag to the benchmark scripts.

The usual approach to such benchmarking, using `valgrind`'s `massif`, necessarily includes measuring the set-up code. As this includes loading both the Python interpreter and PyTorch, measuring the memory usage of our code becomes tricky.

As such we use an alternate method, in which the memory usage is sampled at intervals, using the Python package `memory_profiler`, which may be installed via `pip install memory_profiler`. This in turn has the limitation that it may miss a peak in memory usage; for small calculations it may miss the entire calculation. Furthermore, the values reported are inconsistent with those reported in Reizenstein & Graham (2018).

Nonetheless, when compared against `iisignature` using `memory_profiler`, on larger computations where peaks are less likely to go unobserved, then Signatory typically uses at an order of magnitude less memory. However due to the limitations above, we have chosen not report quantitative memory benchmarks here.

## D.3 SIGNATURE TRANSFORM BENCHMARKS

The precise values of the points of Figures 1 and 2 are shown in Tables 1–4.

For convenience, the ratio between the speed of Signatory and the speed of `iisignature` is also shown.

## D.4 LOGSIGNATURE TRANSFORM BENCHMARKS

See Figure 4 for the graphs of the benchmarks for the logsignature transform.

The computer and runtime environment used was as described in Section 6.

We observe similar behaviour to the benchmarks for the signature transform. `iisignature` is slightly faster for some very small computations, but that as problem size increases, Signatory swiftly overtakes `iisignature`, and is orders of magnitude faster for larger computations.

The precise values of the points on these graphs are shown in Tables 5–8. Times are given in seconds. Also shown is the ratio between the speed of Signatory and the speed of `iisignature`. A dash indicates that `esig` does not support that operation.

## D.5 SINGLE-ELEMENT-BATCH BENCHMARKS

The benchmarks so far considered were for a batch of samples (of size 32). Whilst this is of particular relevance for training, it is sometimes less relevant for inference. We now repeat all the previous benchmarks (forward and backward through both signature and logsignature, varying both depth and channels), except that the batch dimension is reduced to size 1. See Figures 5 and 6. Numerical values are presented in Tables 9–16.

Here we see on very small problems that `iisignature` now outperforms Signatory by about a millisecond, but that once again Signatory overtakes `iisignature` on reasonably-sized problems, and is still orders of magnitude faster on larger problems.

We do not regard the performance on very small single-element problems as a drawback of Signatory. If performing very few very small calculations, then the difference of a millisecond is irrelevant. If performing very many very small calculations, then these can typically be batched together.

Table 1: Signature forward, varying channels. Times are given in seconds. A dash indicates that `esig` does not support that operation.

| Channels | 2 | 3 | 4 | 5 | 6 | 7 |
|---|---|---|---|---|---|---|
| esig | 0.531 | 9.34 | - | - | - | - |
| iisignature | 0.00775 | 0.0632 | 0.375 | 1.97 | 7.19 | 20.9 |
| Signatory CPU (no parallel) | 0.00327 | 0.0198 | 0.101 | 0.402 | 1.45 | 3.8 |
| Signatory CPU (parallel) | 0.00286 | 0.00504 | 0.00975 | 0.0577 | 0.21 | 1.22 |
| Signatory GPU | 0.0129 | 0.0135 | 0.0182 | 0.0222 | 0.0599 | 0.158 |
| Ratio CPU (no parallel) | 2.37 | 3.19 | 3.71 | 4.89 | 4.95 | 5.49 |
| Ratio CPU (parallel) | 2.71 | 12.5 | 38.5 | 34.1 | 34.2 | 17.0 |
| Ratio GPU | 0.602 | 4.68 | 20.6 | 88.7 | 120 | 132 |

Table 2: Signature backward, varying channels. Times are given in seconds. A dash indicates that `esig` does not support that operation.

| Channels | 2 | 3 | 4 | 5 | 6 | 7 |
|---|---|---|---|---|---|---|
| esig | - | - | - | - | - | - |
| iisignature | 0.026 | 0.248 | 1.59 | 7.78 | 27.6 | 128 |
| Signatory CPU (no parallel) | 0.0222 | 0.106 | 0.428 | 1.54 | 4.97 | 13.7 |
| Signatory CPU (parallel) | 0.00922 | 0.0623 | 0.265 | 1.01 | 3.49 | 9.0 |
| Signatory GPU | 0.0472 | 0.0413 | 0.0534 | 0.119 | 0.314 | 0.772 |
| Ratio CPU (no parallel) | 1.17 | 2.34 | 3.7 | 5.07 | 5.56 | 9.38 |
| Ratio CPU (parallel) | 2.82 | 3.97 | 6.0 | 7.69 | 7.92 | 14.2 |
| Ratio GPU | 0.551 | 6.0 | 29.7 | 65.2 | 87.9 | 166 |

Table 3: Signature forward, varying depths. Times are given in seconds. A dash indicates that `esig` does not support that operation.

| Depth | 2 | 3 | 4 | 5 | 6 | 7 | 8 | 9 |
|---|---|---|---|---|---|---|---|---|
| esig | 0.019 | 0.0954 | 0.527 | 2.73 | 15.5 | - | - | - |
| iisignature | 0.000468 | 0.00145 | 0.00485 | 0.0199 | 0.0859 | 0.376 | 1.83 | 8.16 |
| Signatory CPU (no parallel) | 0.000708 | 0.00129 | 0.0022 | 0.00765 | 0.027 | 0.104 | 0.402 | 1.68 |
| Signatory CPU (parallel) | 0.000722 | 0.00242 | 0.00279 | 0.00321 | 0.00546 | 0.0161 | 0.0408 | 0.381 |
| Signatory GPU | 0.00172 | 0.00326 | 0.00484 | 0.00735 | 0.0104 | 0.0132 | 0.0232 | 0.0773 |
| Ratio CPU (no parallel) | 0.661 | 1.12 | 2.2 | 2.61 | 3.18 | 3.6 | 4.55 | 4.86 |
| Ratio CPU (parallel) | 0.649 | 0.597 | 1.74 | 6.21 | 15.7 | 23.3 | 44.8 | 21.4 |
| Ratio GPU | 0.273 | 0.443 | 1.0 | 2.71 | 8.24 | 28.3 | 79.0 | 106 |

Table 4: Signature backward, varying depths. Times are given in seconds. A dash indicates that `esig` does not support that operation.

| Depth | 2 | 3 | 4 | 5 | 6 | 7 | 8 | 9 |
|---|---|---|---|---|---|---|---|---|
| `esig` | - | - | - | - | - | - | - | - |
| `iisignature` | 0.00149 | 0.00438 | 0.0179 | 0.0954 | 0.366 | 1.59 | 7.72 | 34.7 |
| Signatory CPU (no parallel) | 0.00322 | 0.00518 | 0.0123 | 0.0347 | 0.109 | 0.437 | 1.8 | 6.31 |
| Signatory CPU (parallel) | 0.00354 | 0.00409 | 0.0089 | 0.0152 | 0.0563 | 0.175 | 0.839 | 4.06 |
| Signatory GPU | 0.00525 | 0.00916 | 0.015 | 0.0216 | 0.0324 | 0.05 | 0.144 | 0.495 |
| Ratio CPU (no parallel) | 0.464 | 0.845 | 1.45 | 2.75 | 3.37 | 3.63 | 4.28 | 5.49 |
| Ratio CPU (parallel) | 0.422 | 1.07 | 2.02 | 6.28 | 6.51 | 9.07 | 9.21 | 8.54 |
| Ratio GPU | 0.284 | 0.478 | 1.19 | 4.42 | 11.3 | 31.7 | 53.8 | 70.1 |

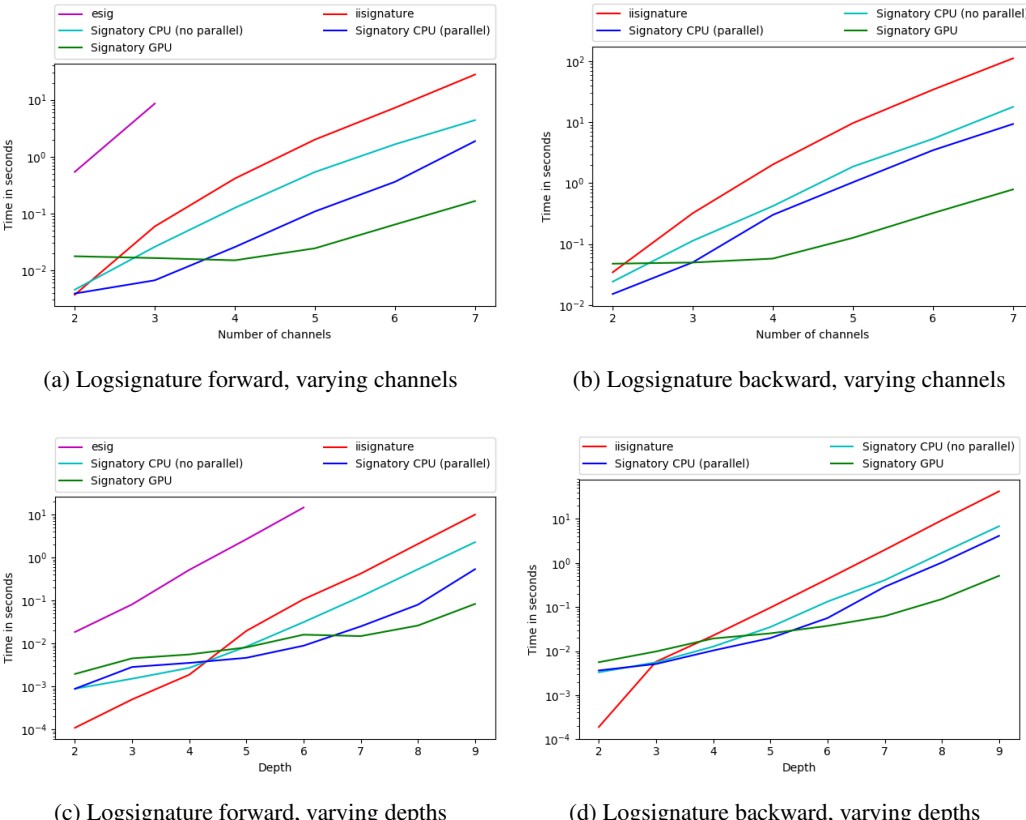

(a) Logsignature forward, varying channels

(b) Logsignature backward, varying channels

(c) Logsignature forward, varying depths

(d) Logsignature backward, varying depths

Figure 4: Time taken on benchmark computations to compute the specified operation. In all cases the input was a batch of 32 sequences of data, each of length 128. For varying channels, the depth was fixed at 7. For varying depths, the channels was fixed at 4. Every test case was repeated 50 times and the fastest time taken. Note that `esig` is only shown for certain operations as it is incapable of computing large operations or of computing backward operations. Note the logarithmic scale.

Table 5: Logsignature forward, varying channels. Times are given in seconds. A dash indicates that `esig` does not support that operation.

| Channels | 2 | 3 | 4 | 5 | 6 | 7 |
|---|---|---|---|---|---|---|
| esig | 0.539 | 8.61 | - | - | - | - |
| iisignature | 0.0037 | 0.0591 | 0.412 | 2.0 | 7.26 | 27.9 |
| Signatory CPU (no parallel) | 0.00454 | 0.0258 | 0.125 | 0.536 | 1.65 | 4.4 |
| Signatory CPU (parallel) | 0.00388 | 0.00665 | 0.0256 | 0.108 | 0.36 | 1.87 |
| Signatory GPU | 0.0176 | 0.0164 | 0.0149 | 0.0243 | 0.0638 | 0.165 |
| Ratio CPU (no parallel) | 0.815 | 2.29 | 3.28 | 3.73 | 4.4 | 6.36 |
| Ratio CPU (parallel) | 0.952 | 8.89 | 16.1 | 18.4 | 20.1 | 14.9 |
| Ratio GPU | 0.21 | 3.6 | 27.5 | 82.2 | 114 | 169 |

Table 6: Logsignature backward, varying channels. Times are given in seconds. A dash indicates that `esig` does not support that operation.

| Channels | 2 | 3 | 4 | 5 | 6 | 7 |
|---|---|---|---|---|---|---|
| esig | - | - | - | - | - | - |
| iisignature | 0.0346 | 0.322 | 2.02 | 9.66 | 34.1 | 111 |
| Signatory CPU (no parallel) | 0.0243 | 0.114 | 0.421 | 1.87 | 5.34 | 17.9 |
| Signatory CPU (parallel) | 0.0152 | 0.0503 | 0.302 | 1.03 | 3.47 | 9.34 |
| Signatory GPU | 0.0478 | 0.05 | 0.058 | 0.127 | 0.323 | 0.79 |
| Ratio CPU (no parallel) | 1.42 | 2.83 | 4.81 | 5.17 | 6.4 | 6.24 |
| Ratio CPU (parallel) | 2.27 | 6.4 | 6.69 | 9.34 | 9.85 | 11.9 |
| Ratio GPU | 0.723 | 6.44 | 34.9 | 76.2 | 106 | 141 |

Table 7: Logsignature forward, varying depths. Times are given in seconds. A dash indicates that `esig` does not support that operation.

| Depth | 2 | 3 | 4 | 5 | 6 | 7 | 8 | 9 |
|---|---|---|---|---|---|---|---|---|
| esig | 0.0185 | 0.0815 | 0.517 | 2.67 | 14.6 | - | - | - |
| iisignature | 0.00011 | 0.000502 | 0.00188 | 0.0197 | 0.107 | 0.423 | 2.07 | 9.98 |
| Signatory CPU (no parallel) | 0.000888 | 0.00152 | 0.00272 | 0.00849 | 0.0315 | 0.124 | 0.534 | 2.28 |
| Signatory CPU (parallel) | 0.000886 | 0.00285 | 0.00355 | 0.00466 | 0.00891 | 0.0251 | 0.0801 | 0.536 |
| Signatory GPU | 0.00196 | 0.00453 | 0.00558 | 0.00815 | 0.0161 | 0.0149 | 0.0262 | 0.0834 |
| Ratio CPU (no parallel) | 0.124 | 0.33 | 0.693 | 2.33 | 3.41 | 3.42 | 3.88 | 4.37 |
| Ratio CPU (parallel) | 0.124 | 0.176 | 0.531 | 4.23 | 12.0 | 16.9 | 25.9 | 18.6 |
| Ratio GPU | 0.0561 | 0.111 | 0.337 | 2.42 | 6.67 | 28.4 | 79.0 | 120 |

Table 8: Logsignature backward, varying depths. Times are given in seconds. A dash indicates that `esig` does not support that operation.

| Depth | 2 | 3 | 4 | 5 | 6 | 7 | 8 | 9 |
|---|---|---|---|---|---|---|---|---|
| `esig` | - | - | - | - | - | - | - | - |
| `iisignature` | 0.000189 | 0.00572 | 0.0225 | 0.0968 | 0.432 | 1.98 | 9.36 | 42.3 |
| Signatory CPU (no parallel) | 0.0033 | 0.00555 | 0.0127 | 0.0351 | 0.133 | 0.408 | 1.69 | 6.84 |
| Signatory CPU (parallel) | 0.00363 | 0.0051 | 0.0103 | 0.0197 | 0.0562 | 0.287 | 1.02 | 4.13 |
| Signatory GPU | 0.00559 | 0.00979 | 0.0193 | 0.0253 | 0.0373 | 0.0621 | 0.151 | 0.511 |
| Ratio CPU (no parallel) | 0.0573 | 1.03 | 1.78 | 2.76 | 3.25 | 4.86 | 5.54 | 6.19 |
| Ratio CPU (parallel) | 0.0521 | 1.12 | 2.18 | 4.91 | 7.69 | 6.89 | 9.13 | 10.2 |
| Ratio GPU | 0.0338 | 0.584 | 1.17 | 3.83 | 11.6 | 31.9 | 61.8 | 82.8 |

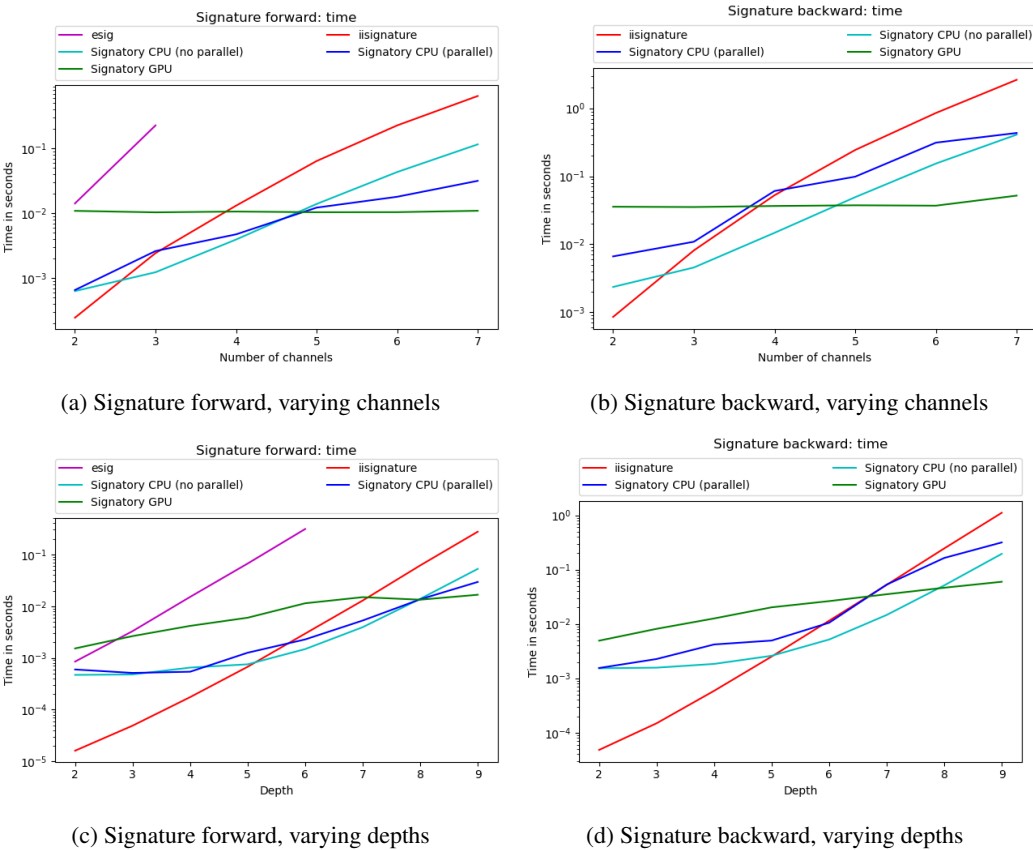

(a) Signature forward, varying channels      (b) Signature backward, varying channels

(c) Signature forward, varying depths      (d) Signature backward, varying depths

Figure 5: Time taken on benchmark computations to compute the specified operation. In all cases the input was a "batch" of 1 sequence, of length 128. For varying channels, the depth was fixed at 7. For varying depths, the channels was fixed at 4. Every test case was repeated 50 times and the fastest time taken. Note that `esig` is only shown for certain operations as it is incapable of computing large operations or of computing backward operations. Note the logarithmic scale.

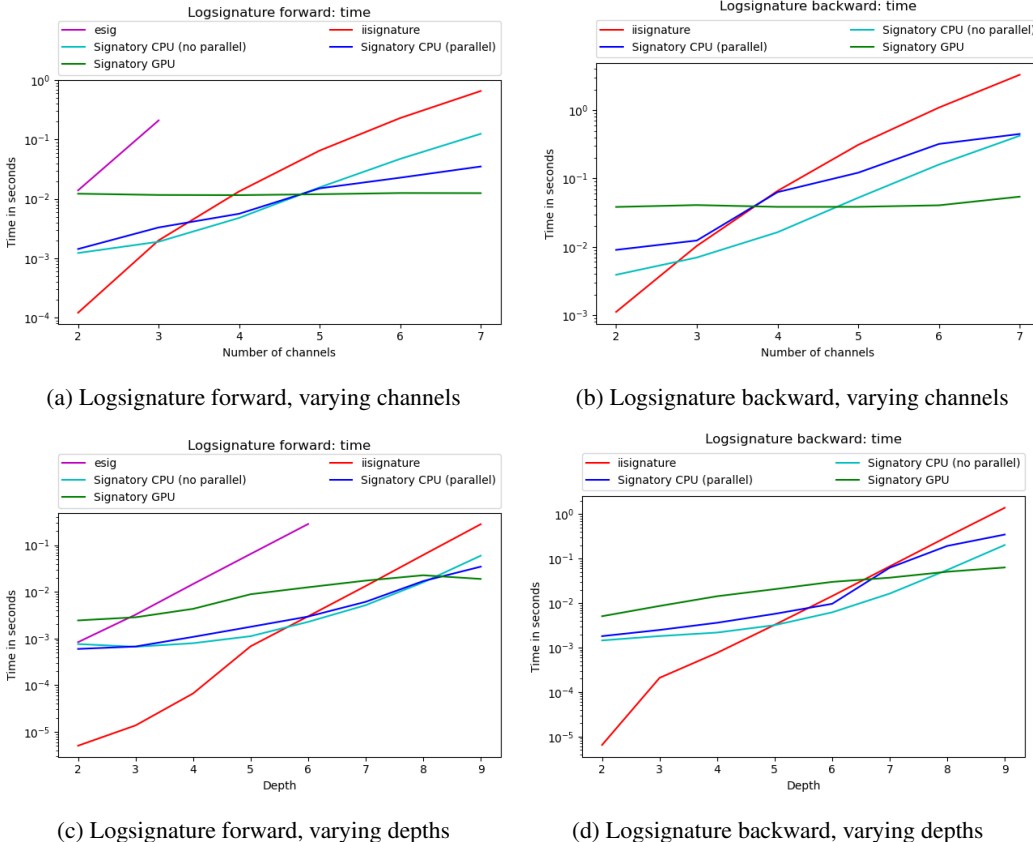

(a) Logsignature forward, varying channels

(b) Logsignature backward, varying channels

(c) Logsignature forward, varying depths

(d) Logsignature backward, varying depths

Figure 6: Time taken on benchmark computations to compute the specified operation. In all cases the input was a "batch" of 1 sequence, of length 128. For varying channels, the depth was fixed at 7. For varying depths, the channels was fixed at 4. Every test case was repeated 50 times and the fastest time taken. Note that `esig` is only shown for certain operations as it is incapable of computing large operations or of computing backward operations. Note the logarithmic scale.

Table 9: Signature forward, varying channels, single-element-batch. Times are given in seconds. A dash indicates that `esig` does not support that operation.

| Channels | 2 | 3 | 4 | 5 | 6 | 7 |
|---|---|---|---|---|---|---|
| esig | 0.0142 | 0.228 | - | - | - | - |
| iisignature | 0.000246 | 0.00244 | 0.0132 | 0.0642 | 0.228 | 0.647 |
| Signatory CPU (no parallel) | 0.000632 | 0.00123 | 0.00394 | 0.0140 | 0.0434 | 0.116 |
| Signatory CPU (parallel) | 0.000657 | 0.00262 | 0.00474 | 0.0123 | 0.0180 | 0.0317 |
| Signatory GPU | 0.0109 | 0.0104 | 0.0107 | 0.0104 | 0.0104 | 0.0110 |
| Ratio CPU (no parallel) | 0.389 | 1.97 | 3.33 | 4.60 | 5.24 | 5.58 |
| Ratio CPU (parallel) | 0.374 | 0.929 | 2.77 | 5.24 | 12.6 | 20.4 |
| Ratio GPU | 0.0225 | 0.235 | 1.23 | 6.17 | 21.8 | 59.0 |

Table 10: Signature backward, varying channels, single-element-batch. Times are given in seconds. A dash indicates that `esig` does not support that operation.

| Channels | 2 | 3 | 4 | 5 | 6 | 7 |
|---|---|---|---|---|---|---|
| esig | - | - | - | - | - | - |
| iisignature | 0.00085 | 0.00809 | 0.0526 | 0.244 | 0.854 | 2.63 |
| Signatory CPU (no parallel) | 0.00235 | 0.00454 | 0.0147 | 0.0492 | 0.154 | 0.41 |
| Signatory CPU (parallel) | 0.00661 | 0.0109 | 0.0607 | 0.0986 | 0.312 | 0.433 |
| Signatory GPU | 0.0356 | 0.0352 | 0.0365 | 0.0374 | 0.0369 | 0.0518 |
| Ratio CPU (no parallel) | 0.362 | 1.78 | 3.58 | 4.95 | 5.55 | 6.41 |
| Ratio CPU (parallel) | 0.129 | 0.745 | 0.867 | 2.47 | 2.73 | 6.06 |
| Ratio GPU | 0.0239 | 0.23 | 1.44 | 6.52 | 23.1 | 50.7 |

Table 11: Signature forward, varying depths, single-element-batch. Times are given in seconds. A dash indicates that `esig` does not support that operation.

| Depth | 2 | 3 | 4 | 5 | 6 | 7 | 8 | 9 |
|---|---|---|---|---|---|---|---|---|
| esig | 0.00085 | 0.0033 | 0.0151 | 0.0668 | 0.311 | - | - | - |
| iisignature | 0.0000161 | 0.0000490 | 0.000175 | 0.000674 | 0.00295 | 0.0129 | 0.0614 | 0.276 |
| Signatory CPU (no parallel) | 0.000470 | 0.000479 | 0.000650 | 0.000753 | 0.00148 | 0.00393 | 0.0139 | 0.0526 |
| Signatory CPU (parallel) | 0.000597 | 0.000512 | 0.000541 | 0.00126 | 0.00228 | 0.00528 | 0.0136 | 0.0294 |
| Signatory GPU | 0.00153 | 0.00264 | 0.00416 | 0.00599 | 0.0114 | 0.0149 | 0.0133 | 0.0166 |
| Ratio CPU (no parallel) | 0.0342 | 0.102 | 0.269 | 0.896 | 1.99 | 3.28 | 4.43 | 5.24 |
| Ratio CPU (parallel) | 0.0269 | 0.0957 | 0.323 | 0.535 | 1.30 | 2.43 | 4.52 | 9.37 |
| Ratio GPU | 0.0105 | 0.0186 | 0.0419 | 0.133 | 0.259 | 0.861 | 4.60 | 16.6 |

Table 12: Signature backward, varying depths, single-element-batch. Times are given in seconds. A dash indicates that `esig` does not support that operation.

| Depth | 2 | 3 | 4 | 5 | 6 | 7 | 8 | 9 |
|---|---|---|---|---|---|---|---|---|
| esig | - | - | - | - | - | - | - | - |
| iisignature | 0.0000480 | 0.000148 | 0.000588 | 0.00251 | 0.0115 | 0.0525 | 0.245 | 1.11 |
| Signatory CPU (no parallel) | 0.00153 | 0.00157 | 0.00184 | 0.00259 | 0.00517 | 0.0146 | 0.0513 | 0.195 |
| Signatory CPU (parallel) | 0.00154 | 0.00226 | 0.00419 | 0.00495 | 0.0105 | 0.053 | 0.164 | 0.317 |
| Signatory GPU | 0.00492 | 0.00813 | 0.0126 | 0.0202 | 0.0263 | 0.0352 | 0.0464 | 0.0598 |
| Ratio CPU (no parallel) | 0.0314 | 0.0947 | 0.320 | 0.969 | 2.22 | 3.58 | 4.79 | 5.70 |
| Ratio CPU (parallel) | 0.0311 | 0.0656 | 0.140 | 0.508 | 1.09 | 0.99 | 1.50 | 3.51 |
| Ratio GPU | 0.00975 | 0.0183 | 0.0465 | 0.124 | 0.436 | 1.49 | 5.29 | 18.6 |

Table 13: Logsignature forward, varying channels, single-element-batch. Times are given in seconds. A dash indicates that `esig` does not support that operation.

| Channels | 2 | 3 | 4 | 5 | 6 | 7 |
|---|---|---|---|---|---|---|
| `esig` | 0.014 | 0.210 | - | - | - | - |
| `iisignature` | 0.000121 | 0.00201 | 0.0134 | 0.0653 | 0.231 | 0.658 |
| Signatory CPU (no parallel) | 0.00123 | 0.0019 | 0.00479 | 0.0157 | 0.0473 | 0.125 |
| Signatory CPU (parallel) | 0.00144 | 0.00331 | 0.00563 | 0.0151 | 0.0228 | 0.0352 |
| Signatory GPU | 0.0123 | 0.0117 | 0.0116 | 0.0120 | 0.0126 | 0.0125 |
| Ratio CPU (no parallel) | 0.0989 | 1.05 | 2.79 | 4.16 | 4.87 | 5.26 |
| Ratio CPU (parallel) | 0.0846 | 0.606 | 2.38 | 4.32 | 10.1 | 18.7 |
| Ratio GPU | 0.00991 | 0.172 | 1.15 | 5.43 | 18.3 | 52.5 |

Table 14: Logsignature backward, varying channels, single-element-batch. Times are given in seconds. A dash indicates that `esig` does not support that operation.

| Channels | 2 | 3 | 4 | 5 | 6 | 7 |
|---|---|---|---|---|---|---|
| `esig` | - | - | - | - | - | - |
| `iisignature` | 0.00112 | 0.0103 | 0.0658 | 0.311 | 1.09 | 3.29 |
| Signatory CPU (no parallel) | 0.00390 | 0.00698 | 0.0163 | 0.0523 | 0.160 | 0.422 |
| Signatory CPU (parallel) | 0.00902 | 0.0124 | 0.0631 | 0.122 | 0.32 | 0.448 |
| Signatory GPU | 0.0383 | 0.0408 | 0.0384 | 0.0385 | 0.0405 | 0.0542 |
| Ratio CPU (no parallel) | 0.286 | 1.48 | 4.02 | 5.95 | 6.82 | 7.81 |
| Ratio CPU (parallel) | 0.124 | 0.834 | 1.04 | 2.56 | 3.41 | 7.35 |
| Ratio GPU | 0.0291 | 0.253 | 1.71 | 8.08 | 26.9 | 60.8 |

Table 15: Logsignature forward, varying depths, single-element-batch. Times are given in seconds. A dash indicates that `esig` does not support that operation.

| Depth | 2 | 3 | 4 | 5 | 6 | 7 | 8 | 9 |
|---|---|---|---|---|---|---|---|---|
| `esig` | 0.000837 | 0.00325 | 0.0148 | 0.065 | 0.285 | - | - | - |
| `iisignature` | 0.00000505 | 0.0000137 | 0.0000666 | 0.000682 | 0.00300 | 0.0134 | 0.0618 | 0.282 |
| Signatory CPU (no parallel) | 0.000762 | 0.000666 | 0.000794 | 0.00112 | 0.00228 | 0.00519 | 0.0160 | 0.0596 |
| Signatory CPU (parallel) | 0.000599 | 0.000674 | 0.00108 | 0.00178 | 0.00296 | 0.00612 | 0.0170 | 0.0347 |
| Signatory GPU | 0.00244 | 0.00285 | 0.00433 | 0.0089 | 0.0126 | 0.0174 | 0.0227 | 0.0189 |
| Ratio CPU (no parallel) | 0.00663 | 0.0207 | 0.0838 | 0.608 | 1.32 | 2.58 | 3.87 | 4.73 |
| Ratio CPU (parallel) | 0.00843 | 0.0204 | 0.0614 | 0.382 | 1.01 | 2.18 | 3.63 | 8.13 |
| Ratio GPU | 0.00206 | 0.00481 | 0.0154 | 0.0766 | 0.239 | 0.767 | 2.72 | 14.9 |

Table 16: Logsignature backward, varying depths, single-element-batch. Times are given in seconds. A dash indicates that `esig` does not support that operation.

| Depth | 2 | 3 | 4 | 5 | 6 | 7 | 8 | 9 |
|---|---|---|---|---|---|---|---|---|
| `esig` | - | - | - | - | - | - | - | - |
| `iisignature` | 0.00000651 | 0.000210 | 0.000765 | 0.00325 | 0.0144 | 0.0670 | 0.311 | 1.39 |
| Signatory CPU (no parallel) | 0.00145 | 0.00182 | 0.00218 | 0.0032 | 0.00622 | 0.0164 | 0.0557 | 0.203 |
| Signatory CPU (parallel) | 0.00182 | 0.00250 | 0.00363 | 0.00575 | 0.00968 | 0.0626 | 0.193 | 0.349 |
| Signatory GPU | 0.00509 | 0.00866 | 0.0143 | 0.0206 | 0.0301 | 0.0374 | 0.0509 | 0.0635 |
| Ratio CPU (no parallel) | 0.00448 | 0.115 | 0.35 | 1.02 | 2.32 | 4.08 | 5.58 | 6.87 |
| Ratio CPU (parallel) | 0.00359 | 0.0839 | 0.211 | 0.566 | 1.49 | 1.07 | 1.61 | 3.99 |
| Ratio GPU | 0.00128 | 0.0242 | 0.0534 | 0.158 | 0.48 | 1.79 | 6.11 | 21.9 |

