# OpenReview forum: "Signatory: differentiable computations of the signature and logsignature transforms, on both CPU and GPU"
_ICLR.cc/2021/Conference — ICLR 2021 Poster_

### Official Review · AnonReviewer2 · 2020-10-28
**Algorithms for calculating path signatures**

**Rating:** 7
**Confidence:** 3

**Review:**

Signatory is a library for calculating path 'signatures'. Unlike previous implementations, it engineered with machine learning uses cases in mind, supporting minibatches and GPU computation. The performance seems to be a significant improvement on previous work. This seems to be stem from a mix of algorithmic modifications and engineering work.

#### Questions/Comments:

In it unclear how much of the the performance improvement is due to (i) technical optimizations like using AVX to speed up batch processing, and (ii) due to fundamental algorithmic improvements, e.g. the choice of basis.
* How does performance compare with esig/iisignature in the batch size == 1 case, where using AVX is harder? Although batching is common during training, the 'batch size = 1' case is important in applications.
*  Figures 5 and 6 in [A] show the FLOPs needed for each step of the path for specific dimensions and depths.
Can you compare the number of FLOPs per piecewise-linear-segment needed by Signatory and iisignature to calculate the  log signature path-increment update?

p4:Section 3.1: Is the fusion a contribution of the paper, or is it similar to previous work, i.e. Section 4 in [A]? Is the difference that Signatory does kernel fusion for the signature and log-signature, but iisignature only does it for the signature?

p5: 'numerical stability' The experiments in article look at the run time, but not the numerical accuracy.
This raises a theoretical concern that Signatory might be less numerically stable than previous libraries.
(C.f. In the case of matrix multiplication, Strassen's algorithm uses fewer FLOPs than the naive method, but is less numerically stable.) Is Signatory 'fit for purpose' for machine learning in the sense that either
(i) the calculated signatures elements are within 5% of the true values, or
(ii) when used in a machine learning context, test accuracy is equivalent to previous libraries?


#### References:
[A] Reizenstein and Graham, https://arxiv.org/pdf/1802.08252.pdf

----updating in light of author's response-----
I am upgrading to 7 as my concerns have been addressed.
Other reviewers have questioned if ICLR is the right venue for a paper about a library.
The paper does describe novel algorithms that lower the big-O cost of computation, similar to how Strassen's algorithm is a non-trivial modification of the naive matrix multiplication algorithm.

---

> ### Author Response · Authors · 2020-11-12
> **Questions answered; new benchmarks available.**
>
> Thankyou for your review.
>
> In response to each comment, in order:
>
> **Batch size = 1**
> We have added additional benchmarks to the appendix for the batch size = 1 case. In these additional benchmarks, iisignature gains some ground for the absolute smallest of computations, which we attribute to various small overheads - additional argument checking, unwrapping PyTorch, dispatching to the optimal implementation, etc. Signatory maintains its good performance.
>
> Incidentally, during initial construction of the libary, we actually found naive parallelism over the batch dimension to be slightly superior to vectorisation, although this was marginal.
>
> Relatedly, we have also added an additional benchmark to the main paper, comparing Signatory against iisignature on a "typical" deep learning problem; we find that the model using Signatory train 210 times faster than the one using iisignature.
>
> **FLOPs**
> A comparison of the number of multiplications for an update over a single linear segment is already included in Appendix A, which we believe demonstrates the essential point.
>
> **Fusion**
> The fusion is indeed a contribution of the paper. Section 4 of [A] is different. Our fused operation is a _mathematical_ improvement to computing the _signature_; Section 4 of [A] discusses a _software_ improvement for computing the _logsignature_.
>
> **Numerical stability**
> As the fused operation relies on an expansion in the style of Horner's method then in fact we expect it to be more numerical stable than previous approaches. This is practically validated: Signatory's test suite validates its results against that produced by iisignature.
>
> **Summary**
> We hope this has addressed the reviewer's concerns. If there any additional improvements that can be made to the paper so as to improve the review rating, then we would be happy to discuss these too.

---

> > ### Comment · AnonReviewer2 · 2020-11-19
> > **FLOPs**
> >
> > **Batch-size = 1**
> > Thank you for the new Figures 5&6 (p. 23-24). So signatory is between 100 slower and 10 times faster for batch size 1.
> > Given that you claim to have a superior algorithm, I am still curious about how the number of FLOPs compare between the two methods.
> >
> > **Flops and fusion**
> > I don't follow your answer. Suppose I just want to find the signature or log-signature (but I don't care which) in as few calculations as possible.  Section 4 of [A] use the Baker–Campbell–Hausdorff formula to simplify the line-segment concatenation operation.
> > Are you saying that that is the same as 'The conventional way' discussed in appendix A?

---

> > > ### Author Response · Authors · 2020-11-19
> > > **Response**
> > >
> > > We'll answer these points the other way around for clarity.
> > >
> > > **FLOPs and fusion:** Specifically consider the signature. The previous algorithm for doing this ("the conventional way"), and the new one we develop, are what are analysed in Appendix A. In particular, we show that the new algorithm requires fewer operations, that it has lower asymptotic complexity, and that this asymptotic complexity is now optimal.
> > >
> > > **Batch size = 1:**  FLOPs are as addressed in the previous section. Additionally, we note that either:
> > > - A single small operation is required -- in which case differences of milliseconds are usually irrelevant;
> > > - Many small operations are required, in which case batching may be used -- and Signatory is the clear leader;
> > > - The operation is large -- in which case Signatory is again the clear leader.
> > >
> > > It is for this reason that multiplicative descriptions (e.g. "100 times slower") are typically less relevant for single very small operations.
> > >
> > > Is this clear / do you agree?

---

> > > > ### Comment · AnonReviewer2 · 2020-11-23
> > > > **FLOPs**
> > > >
> > > > Thanks for the reply.
> > > > I remain curious to see how your method compares with [A] in terms of the number of FLOPs. Batching is relevant at training time, but a successful AI system should spend the bulk of its time in the testing/use stage where batching is less practical. As it is, it is hard to evaluate the claim in Section 4.3 that you have a better basis.
> > > >
> > > > To pick one example, figure 6 in [A] shows that 24 FLOPs (fused multipy-add operations) are needed to add a line segment in the dimension=2, depth=3 case. What is the equivalent number for your algorithm?

---

> > > > > ### Author Response · Authors · 2020-11-23
> > > > > **20 FLOPs**
> > > > >
> > > > > Quoting from equation (11) of the paper, our algorithm uses $\mathcal{F}(d=2, N=3) = 20$ operations.

---

### Official Review · AnonReviewer1 · 2020-10-28
**Good implementation of an interesting and promising transformation**

**Rating:** 8
**Confidence:** 3

**Review:**

Summary
-------
The paper presents the first GPU-capable library implementing the _"signature"_ and _"log-signature"_ functions as well as their gradients. It introduces these transformations to a machine learning audience, as well as their recent uses in ML, then proposes algorithmic improvements that reduce the necessary computation. The resulting library is benchmarked against existing implementations, and the code, benchmarks, and proofs are included in supplementary materials.

Pros
-----
* The library makes the signature transform much more accessible to ML researchers, opening up a promising research space
* The paper is overall really clear, especially the introduction to the signature function and the explanations about the implementation strategy
* Benchmarks are sound and show impressive speed-ups; they seem particularly easy to reproduce thanks to the detailed instructions
* The work goes beyond implementation of an existing algorithm in a new framework, but uses novel techniques to reduce algorithmic complexity

Cons
------
* Very few details about the implementation of back-propagation through these transforms, and of the inverse transforms, although they are an important part of the implementation

Recommendation
----------------------------
I recommend to **accept** this paper, as the proposed library could be extremely helpful to researchers wanting to explore the signature function, especially as an intermediate transformation. The impact could be high in sequence modelling for instance.

Arguments
------------------
* A fast, differentiable implementation of an operation of interest, especially integrated in a major framework, is *highly significant* for the community, as it enables quick exploration of research ideas to incorporate it.
* The paper is *clearly written*, and does a good job exposing the concepts of interest to an unfamiliar audience. Relevant literature is clearly cited with context.
* The *quality* of the benchmarks is god, and care has been put in making them easily reproducible. The algorithmic improvements (reformulation of complex formulas to re-use common computations, pre-computation and re-use of inverses) are clever and especially adapted to the current ML/DL context, in particular getting rid of a costly linear transformation in the log-signature computation.
* The work is *original* in that it is the first library to include both GPU compatibility and implementation of the gradients of such transformations, as well as the speed ups mentioned above.

Clarifications
------------------
The main missing part is the derivation and implementation of the gradients of these transformations for reverse-mode automatic differentiation. The text glosses over it in section 4.4, but I would have appreciated if its implementation had been explained, similarly to the original transforms. For instance:
- Is the implementation of the gradients similar or dissimilar to the forward computation?
- Are there common parts that are re-used in both?
- Are there different trade-offs?
- Do the implementations benefit from the same algorithmic improvements, or other ones?
- What are the opportunities for parallelism?

From what I understand, the "inverse" transformations are really close to the original ones, but it could have been expanded upon as well.

Minor points:
- In section 4.5.4, the text mentions computing Eq. 7 not by combining intervals (as in 4.5.1), but exploiting the fused multiply-exponentiate. I assumed that section 4.5.2 was already using the fused multiply-exp for expanding intervals, is that not the case? Or is there another difference I missed between the techniques of sections 4.5.2 and 4.5.4?
- In section A.2.1, do I understand correctly that $a_2 a_2$ would _not_ be a Lyndon word? There is no other rotation that comes before it, but it does not come *strictly* before itself either.

Additional feedback
--------------------------------
It would have been particularly nice to have an empirical evaluation of the removal of the new basis of the log-signature function. As the authors state, it is unlikely to have a major impact if a linear transformation is learned afterwards, but there are cases where linear transformations can impact the dynamics of training, by improving or worsening conditioning for instance. I understand it is not the focus of this work, though.

Typos:
- A.1.1: "naiïvely" -> "naïvely"
- A.2.1: In the definition of the longest Lyndon suffix, should it be "the smallest $j > 1$?
- A.2.2: In the unnamed equation before (15), should it be $1 \leq i_1, ...$ ?
- C.1 : "reproducability" -> "reproducibility"

---

> ### Author Response · Authors · 2020-11-12
> **Added new section on backpropagation**
>
> Thankyou for your review.
>
> We are very pleased that the paper is identified as clearly written and of high quality, and that the libary itself is also praised as highly significant.
>
> The reviewer expresses concern that backpropagation is merely glossed over. We agree -- as such we have now added a new section on this topic to the appendix (using the reviewer's suggestions as prompts). Likewise, we have added a small amount of extra discussion of the inverse transforms.
>
> **Regarding the minor points:**
> - Sections 4.5.2 and 4.5.4 are indeed quite similar, and both utilise the fused multiply-exponentiate. The difference is mainly one of functionality (rather than mathematics), in that 4.5.4 starts with an existing signature, whilst 4.5.2 does not, which has implications for computational efficiency.
> - The reviewer is correct, $a_2 a_2$ would not be a Lyndon word. We have added a note on this.
>
> **Regarding the additional feedback:**
> Indeed, linear transformations can occasionally have an impact, batch normalisation being the most famous example, but we agree this is not the focus of this work.
>
> Thankyou additionally for catching several typos even in the supplementary material (and indeed the reviewer is correct regarding the typos in A.2.1 and A.2.2).
>
> **Summary:**
> We hope we have addressed all points of feedback, and have worked to improve the paper in response.

---

### Official Review · AnonReviewer3 · 2020-10-28
**Good content, but requires stronger presentation and framing**

**Rating:** 7
**Confidence:** 3

**Review:**

After reading the rebuttal from the authors, as well as the updated draft, I agree that this style of presentation and framing is much more approachable to people not familiar with sig/logsig transforms. Thus I vote to accept this as a *library* paper.



Library papers are difficult to review, and in general reviewing them is a highly subjective process (far more subjective than reviewing in general). Although I think this library paper could be a good contribution, I'd like a subsequent edit pass by the authors with a strong focus on presentation and framing before I am fully convinced.

Above all, I'd really appreciate a library paper being an *advertisement* for the library. In the current form the paper doesn't do a good job advertising the signature and logsignature transforms as a must have in a researchers toolkit. As this is the first time many readers will learn about the existence of the signature/logsignature transform, I'd like a more significant portion of the paper's content to motivation for why these transforms are important in machine learning, as well as useful applications. What problems become significantly easier due to this transform? What useful properties does it have? I think that this deserves more real estate than say comparisons to competing libraries, as well as the precise mathematical details, which could be moved to the appendix.

It's unclear to me how to best apply this transform just by reading the paper. Interestingly, I found other application papers did a better job of describing the signature/logsignature transform. This is a definite shortcoming of this paper as I would deeply appreciate a 'text' explanation of how these transforms work. Indeed it seems that the strongest motivation for sig/logsig is their apparent purpose as a 'summarizer' of arbitrary time series. To this extent sig/logsig behave as a universal 'sketching'/compressed sensing tool for time series. Yet, I had to read other papers to figure this out.

Outside of a few nitpicks, I found the mathematical presentation and definitions of the paper high quality and illuminating. In particular, after understanding what sig/logsig transform actually do, I could easily match notation to concept. I urge the authors to both present sig/logsig transforms conceptually as well as mathematically and help the reader match concept to notation.

Is the purpose of backprop through sig/logsig transform to allow for usage in larger models? Could the authors give a usecase of this? Using this in practice inside a larger deep learning model would give significant weight to this contribution. Indeed, usage in a wider deep learning application of these transforms with good results would make this library a bit of a 'must-have'.

In Eq. 1 is using \prod to define an iterated cartesian product standard? A footnote here would be useful to make this notation explicit.

Should Sec 2.3 be ahead of 2.2 as it is more conceptually tied to the def'n provided in 2.1.

As the mathematical definition of tensor product is used here, could the authors clarify in the appendix/give pointers in the main paper?

In experiments how come the fastest speed is presented. Why not mean/stderror as usual?

Plenty of space is wasted in Sec. 4 highlighting the contributions of the library. Can this be improved with the use of \paragraphs instead of \subsubsections?

Applications/usecases/toy examples solved by this library would be so useful to present along with benchmarks. How difficult is this library to code with? Does it remain 'pythonic'? Does it blend well with existing pytorch code?

---

> ### Author Response · Authors · 2020-11-12
> **Adverts - agree to disagree? Everything else - done!**
>
> Thankyou for your review.
>
> **On adverts**
> Regarding the central comment that the library paper should be an advert -- unfortunately we have to disagree. As a research article, our primary concern is that this be a _technical description of the library_. This is of particular importance as time, and libraries, progress. We believe the implementer of some future library would probably prefer a description of the functionality and technical considerations that should be borne in mind.
>
> On a practical note, we have already seen great uptake of Signatory within our community -- an advertisement is scarcely necessary.
>
> Similarly, the intended audience is someone already familiar with the topic, and do not intend that this paper be a first introduction. We do appreciate that this is not the experience as a reviewer, though.
>
> **Everything else**
> This concern aside, we are pleased that the presentation is identified as high quality and illuminating, and have sought to implement the reviewer's other suggestions. In the order that they are stated in the review:
>
> We have added a new section giving intuition for the nature of the (log)signature transform. This should help give an overview of how it is used, and give a high-level conceptual overview. Moreover, we have added a new deep learning benchmark to the end, to give an example of how this may be used in models in practice, and to demonstrate a 210 times speed-up over previously available libraries.
>
> On that note, this example also includes backpropagation through the signature, which is indeed of substantial interest. In particular this was discussed in "Deep Signature Transforms" (NeurIPS 2019).  See also Section 1.2 of our paper -- many of the users listed there rely on the availability of backpropagation.
>
> \prod to define an iterated Cartesian product is indeed standard.
>
> Section 2.3 relies on the grouplike structure (the existence of the group inverse) and must therefore come after Section 2.2.
>
> We don't completely understand the concern about the mathematical definition of the tensor product. Could the reviewer expand on this concern please?
>
> When benchmarking deterministic software, it is more typical to provide the fastest speed, not the mean. The reason for this is that the measurement errors are one-sided: it is not possible to obtain measurements that are faster than is possible. Much guidance on this topic can be found online; we refer to e.g. [1].
>
> As suggested we have reorganised "New Features" the section to help make space. Likewise, we appreciate the importance of some conceptual exposition, and have added some.
>
> We agree that a small amount of example code demonstrating the use of the library would be beneficial, and have added this as a new section. It is brief, but hopefully makes clear that Signatory integrates seamlessly with existing PyTorch code. (Thus remaining easy to code with; and be Pythonic.)
>
> **Summary**
> We believe we have addressed all of the reviewer's concerns, and hope that they will raise their review score. We would be very happy to discuss this further if desired.
>
> [1] https://blog.kevmod.com/2016/06/benchmarking-minimum-vs-average/

---

### Official Review · AnonReviewer4 · 2020-10-29
**Software for the signature transform**

**Rating:** 6
**Confidence:** 3

**Review:**

Update:
The authors have revised the paper, which helps the presentation somewhat (though headings like "The Grouplike Structure" still come at the reader without much context).

The authors added a more application-oriented benchmark, which makes the more convincing case for practical speedup of 210x.

Certainly the new "Intuition" section is helpful in explaining the transform.

The NT library is an interesting counterpoint on the library front. It feels a little bit apples and oranges to me because of the broad scope of that library (give me DNN, I give you NTK) as opposed to the narrower scope of this one (give me sequence, I give you [log][invert]signature). NT is munging your entire DNN into a GP kernel; signature is an implementation of O(a dozen) ops.

I remain somewhat skeptical that the signature transform can enjoy wide applicability given the exponential scaling behavior, unless first and perhaps second order terms suffice for practical use-cases.

---

The main contribution is a software library for computing the signature and logsignature transforms taking advantage of CPU parallelism and GPU acceleration, and providing a reverse mode derivative. Some algorithmic improvements which yield substantial speedups to the computation are described. Features of the library are described, offering more functionality than predecessor software packages.

The paper provides a background of the signature and related transforms, motivating these with selected applications including continuous differential equations (relevant to continuous normalizing flows), sepsis prediction, handwriting ID, GANs, etc.

The authors demonstrate in several benchmarks that the new library is substantially more performant (10-100x) than predecessors, which suggests it could unlock further usage by making it more practical / reducing time-per-step.



+ve:
Insofar as "signature" is a representation and this software lets us do backprop, this might be a fit to ICLR.
Or perhaps it fits into the "implementation issues, parallelization, software platforms, hardware" CFP.

As machines become more powerful, we gain the capability of using such representations, and having software to enable this is important.

The transform can be useful in neural differential equations, an area of some interest to the community, though exploring how is beyond the scope of the paper. Stepping back, the transform seems to be used in several areas-of-interest to ICLR.

-ve:
While some math background is given, it feels like 8 pages may be too tight to present it thoroughly. I was not able to understand from this background what the invert/log/invertlog signature transforms are, other than "log is a compressed representation".

This is "just" about the software. It would be feel more compelling to me if the software were offered as the artifact of an application paper, or at least if the benchmarking portion were on an application instead of the pure forward/backward passes of the signature transform. As it stands, we can only speculate that there must be cases where signature was the sole bottleneck and such training setups are now ~10-100x faster. To frame it another way, it would be more interesting to see a plot of wall clock on the x axis and loss on the y axis, for some real world problem where this transform is useful, and note speedups as "reaches loss y 50x faster".

The paper does not discuss what appears to be a substantial downside of the approach, namely exponential scaling of memory in the truncation term N. Memory scaling may be in the appendix. What is the tradeoff between "universally nonlinear" and truncating N to practical values? The $d^N$ term seems quite limiting for multidimensional timeseries even as d goes to 5 or 10.

The actual algorithmic changes are around rearranging multiply-exp, and around recasting a sequential reduction as an associative reduction for parallelism.


Accept/reject
Undecided / weak reject

My main reservations are mainly around the goodness of fit of a solely-software paper to ICLR. At a glance, I would think the paper might better appear in a forum like SysML. Certainly the CFP includes bullets the software is relevant-to; but the way I read the CFP is that we are looking for research innovations in those spaces. Implementing accelerator friendly kernels, rearranging the math, and implementing backprop, while all very valuable contributions in a purely applied sense, do not in themselves seem to match the CFP. My opinion is pretty weak, though, because the huge speedups very well may unlock novel usages; and the "software platforms" part of the CFP could maybe be interpreted to match this work.

---

> ### Author Response · Authors · 2020-11-12
> **Not solely software; feedback implemented**
>
> Thank you for your review. We appreciate software papers are not commonly submitted, so there is some uncertainty about them.
>
> Certainly we believe that this is a good fit for ICLR -- there is precedent. See for example "Neural Tangents: Fast and Easy Infinite Neural Networks in Python" (ICLR 2020), which as a software package in fact received a spotlight at last year's ICLR. (And similar comments about the infrequency of software papers were made then, too.)
>
> We have additionally sought to address the reviewer's other concerns.
> - Regarding the concern that there should be "research innovations", and this is a paper "just about the software" - much of our paper does focus on technical discussion and algorithmic improvements. (Much of it necessarily deferred to the appendices for space.) Moreover we attribute our success over previous offerings to be _because_ of these algorithmic improvements. In summary, this is not a paper solely about software.
> - That it would be compelling to have an applicaiton-based benchmark: we have added such a benchmark, comparing Signatory to iisignature for training a signature-based deep learning model. We find that the model using Signatory trains 210 times faster than iisignature. (And have to provide an additional log-scaled plot to even compare them.)
> - We agree that 8 pages seems too tight to really get into the background mathematics. To help ameliorate this we have added a new section "2.5: Intuition".
> - The exponential scaling is indeed a limiting factor for signature methods. This is actually a well-known and well-studied topic in the community; we have added some references to this.
>
> We hope that this addresses the reviewer's concerns and welcome further discussion towards improving their score.

---

### Decision · Program_Chairs · 2021-01-07
**Final Decision**

**Decision:**

Accept (Poster)

**Comment:**

This paper introduces Signatory, a library for computing functionality related to the signature and logsignature transforms. Although a large body of the initial literature on the signature in ML focuses on using it as a feature extractor, more recent works have incorporated within modern deep learning architectures and therefore, the importance of having GPU-capable libraries (with automatic differentiation) that implement these transforms. Several algorithmic improvements are incorporated into the library. Some of the computational benefits of this library wrt to previous ones are demonstrated empirically.

There were some concerns from the reviewers about accepting library papers at ICLR. Library papers clearly fall into the ICLR CFP and, therefore, library, frameworks and platform papers that can be relevant and impactful are welcome contributions to the community. Additionally, more signature-related papers are appearing are mainstream ML venues, hence, despite the poor scalability wrt input dimensions, this paper is definitely relevant.

Perhaps one the drawbacks of this paper is the lack of a more rigorous empirical evaluation. The authors have added a deep learning benchmark, which is welcome but only on a toy dataset. There are still some concerns about the wide applicability of the signature (and its relatives) given its exponential scaling. That’s why applications on more realistic problems will be welcome. At the very least, It will be good if the authors incorporate a separate section discussing the limitations of the signature transform (and the library), especially in terms of computations and scalability.